# LINEAR BANDITS WITH PARTIALLY OBSERVABLE FEATURES

## ABSTRACT

We introduce a novel linear bandit problem where a subset of features is latent, resulting in partial access to reward information and spurious estimates. Without properly addressing the latent features, the regret grows linearly over the decision epoch $T$ while improving the regret bound is challenging because their dimension and relationship with rewards are not available. We propose a novel analysis to handle the latent features and an algorithm that achieves a regret bound sublinear in $T$. The core of the algorithm lies in (i) augmenting basis vectors orthogonal to the observable feature space, and (ii) developing an efficient doubly robust estimator that further improves the regret bound. With these two ingredients, our algorithm achieves a regret bound of $\widetilde{O}(\sqrt{(d + d_h)T})$, where $d$ is the dimension of observable features, and $d_h$ is the *unknown* dimension of the unobserved features that affects the reward. Crucially, our algorithm does not rely on prior knowledge of the unobserved feature space, which expands as more features become hidden. Numerical experiments confirm that our algorithm outperforms both non-contextual multi-armed bandits and other linear bandit algorithms.

## 1 INTRODUCTION

We consider a linear bandit problem where the learning agent has access to only a *subset* of the features, while the reward is determined using the *complete set* of features, including both observed and unobserved elements. Conventional linear bandit problems rely on the assumption that the rewards are linear to only observed features, without accounting for the potential presence of unobserved features. However, in many real-world applications, rewards are often affected by the latent features that are not observable to the agent. For example, in recommendation systems, the true reward — such as user satisfaction or purchase decisions — depends not only on observable features like user demographics or past behaviors but also on latent preferences, such as specific tastes in artists (for streaming services) or brands (in e-commerce). Accurately incorporating these latent features is essential for providing precise recommendations, while ignoring them causes bias or model misspecification errors in every decision-making.

To address the latent features, Park & Faradonbeh (2022), Kim et al. (2023a) and Park & Faradonbeh (2024) rely on the assumption that observed features are linear to the latent features sampled from a specific distribution, e.g., a mean-zero Gaussian. Establishing a regret bound sublinear in the decision horizon without such structural assumptions on the latent features remains a significant challenge and has not been accomplished yet. Key challenges in the bandit problem with partially observable features arise from the complete lack of information on the latent features. Indeed, we do not even know whether an agent observes features partially or not and whether we should use the latent features or not.

To address these challenges, we propose a novel linear bandit algorithm that is agnostic to the presence of partially observable features. Notwithstanding the absence of knowledge regarding unobserved features, our algorithm is capable of obtaining a regret bound that is tighter than that achieved both linear bandit algorithms that consider only observable features and multi-armed bandit (MAB) algorithms that entirely ignore features. Our proposed algorithm achieves a regret bound of $\widetilde{O}(\sqrt{T})$, without requiring any prior knowledge of the unobserved features, where $T$ is the decision horizon and $\widetilde{O}(\cdot)$ represents Big-O notation omitting logarithmic factors.

Table 1: An overview of regret bound range of our algorithm, RoLF, depending on $d_h \in [0, K-d]$, the dimension of the vector space spanned by the rows of the matrix of unobserved features influencing the reward. Note that $\widetilde{O}$ denotes the big-O notation omitting logarithm factors.

| Feature space | Regret bound |
| --- | --- |
| span(observed features) $\supseteq$ span(latent features) | $\widetilde{O}(\sqrt{dT})$ |
| span(observed features) $\subseteq$ span(latent features) | $\widetilde{O}(\sqrt{KT})$ |
| otherwise | $\widetilde{O}(\sqrt{(d + d_h)T})$ |

The key idea of our proposed algorithm lies in two main components: (i) reconstructing the feature vectors to capture the impact of unobserved features on the rewards, and (ii) constructing a novel doubly robust estimator that is robust to information loss caused by unobservability. For (i), we decompose the rewards into two additive terms: one projected onto the row space of the observable features, and the other onto its orthogonal complement. The former term maximally captures the effects from the observed features, while the latter minimizes the impact of the unobserved features. We then augment the observable features with an orthogonal basis from the complement space to capture all effects on the rewards. This allows us to reformulate the problem in a conventional linear bandit framework, where the reward function is defined as the dot product of the minimally augmented features and the associated unknown parameter. However, these augmented features are not identical to the unobserved features, which may lead to potential estimation error. To mitigate these errors, we leverage (ii) the doubly robust estimator, which is widely used in statistical literature for its robustness to errors caused by missing data. Together, these two approaches allow the algorithm to effectively compensate for missing information, enhancing both estimation accuracy and adaptability to the environment.

Our main contributions are summarized as follows:

- We propose a linear bandit problem with partially observable features. Our problem setting is more general and challenging than those in the existing literature on linear bandits with latent features, which often rely on specific structural assumptions governing the relationship between observable and latent features. In contrast, our approach assumes no additional structure for the unobserved features beyond the linearity of the reward function, which is commonly adopted in the linear bandit literature (Section 3).

- We introduce a novel estimation strategy by (i) efficiently augmenting the features that maximally captures the effect of reward projected onto the observed features, while minimizing the impact of unobserved features (Section 4), and (ii) constructing a doubly-robust (DR) estimator that is robust to the error caused by unobserved features. By integrating augmented features with the DR estimator, we guarantee a convergence rate of $\widetilde{O}(t^{-1/2})$ on the rewards for *all* arms in each round $t$ (Theorem 2).

- We propose the *Robust to Latent Features* (RoLF) algorithm for the general linear bandit framework with latent features that achieves a regret bound of $\widetilde{O}(\sqrt{(d + d_h)T})$ (Theorem 3), where $d_h$ is the dimension of the subspace formed by projecting the reward, linear to unobserved features, onto the orthogonal complement of the row space of the observable features (Section 4.2). Our proposed algorithm requires no prior knowledge or modeling of the unobserved features, yet achieves a sharper regret bound than both linear bandit algorithms that consider only observable features (Li et al., 2010; Abbasi-yadkori et al., 2011; Agrawal & Goyal, 2013; Kim & Paik, 2019) and MAB algorithms (Auer et al., 2002).

- Our numerical experiments demonstrate that our proposed algorithm consistently outperforms the existing linear bandit and MAB algorithms. These results support our theoretical findings and validate the practicality of our method.

## 2 RELATED WORKS

In bandit problems, the learning agent learns only from the outcomes of chosen actions, leaving unchosen alternatives unknown (Robbins, 1952). This constraint requires a balance between exploring

new actions and exploiting actions learned to be good, known as the exploration-exploitation tradeoff. Efficiently managing this tradeoff is crucial for guiding the agent towards the optimal policy. To address this, algorithms based on *optimism in the face of uncertainty* (OFU) (Lai & Robbins, 1985) are widely used and studied in linear bandits (Abe & Long, 1999; Auer, 2002; Dani et al., 2008; Rusmevichientong & Tsitsiklis, 2010). Notable examples include `LinUCB` (Li et al., 2010) and `OFUL` (Abbasi-yadkori et al., 2011), known for their practicality and performance guarantees. However, existing approaches differ from ours in two key aspects: (i) they assume that the learning agent can observe the entire feature vector related to the reward, and (ii) their algorithms have regret that scales linearly with the dimension of the observed feature vector, i.e., $\widetilde{O}(d\sqrt{T})$.

In contrast, we develop an algorithm that achieves a sublinear regret bound by employing the doubly robust (DR) technique, thereby avoiding the linear dependence on the dimension of the feature vectors. The DR estimation in the framework of linear contextual bandits is first introduced by Kim & Paik (2019)andDimakopoulou et al. (2019), and subsequent studies improve the regret bound in this problem setting by a factor of $\sqrt{d}$ (Kim et al., 2021; 2023b). A recent application (Kim et al., 2023c) achieves a regret bound of order $O(\sqrt{dT\log T})$ under IID features over rounds. However, the extension to non-stochastic or non-IID features remains an open question. To address this issue, we develop a novel analysis that applies the DR estimation to non-stochastic features, achieving a regret bound sublinear with respect to the dimension of the augmented feature vectors. Furthermore, we extend DR estimation to handle sparse parameters, thereby further improving the regret bound to be sublinear with respect to the reduced dimension.

Our problem is more general and challenging than misspecified linear bandits, where the assumed reward model fails to accurately reflect the true reward, such as when the true reward function is non-linear (Lattimore & Szepesvári, 2020), or a deviation term is added to the reward model (Ghosh et al., 2017; Bogunovic et al., 2021; He et al., 2022). While our work assumes that the misspecified (or inaccessible) portion of the reward is linearly related to certain unobservable features, misspecified linear bandit problems can be reformulated as a special case of our framework. While the regret bounds in Lattimore & Szepesvári (2020), Bogunovic et al. (2021) and He et al. (2022) incorporate a sum of misspecification errors that may accumulate over the decision horizon, our work establishes a regret bound that is sublinear in the decision horizon $T$ without any misspecification errors. Ghosh et al. (2017) proposed a hypothesis test whether to use linear bandits or MAB and proved $O(K\sqrt{T}\log T)$ regret bound when the sum of misspecified error is greater than $\Omega(d\sqrt{T})$. In contrast, our algorithm attains $O(\sqrt{(d+d_h)T\log T})$ regret bound without necessitating such hypothesis tests for misspecification or partial observability.

Lastly, our problem appears similar to the bandits with partially observable features studied by Park & Faradonbeh (2022). In their work, the observed features are assumed to be related to the latent features through a known linear mapping, with the latent features sampled from a centered Gaussian distribution. However, our approach does not impose any structural assumptions on either the observed or latent features, making it a more general and challenging problem compared to that of Park & Faradonbeh (2022).

## 3 PRELIMINARIES

### 3.1 NOTATION

For any $n \in \mathbb{N}$, let $[n]$ denote the set $\{1, 2, \ldots, n\}$. Furthermore, the $L_1$, $L_2$ and supremum norm of a vector $\mathbf{v}$ is represented by $\|\mathbf{v}\|_1$, $\|\mathbf{v}\|_2$, and $\|\mathbf{v}\|_\infty$, respectively, and the $L_2$-norm weighted by a positive definite matrix $\mathbf{D}$ is denoted by $\|\mathbf{v}\|_{\mathbf{D}}$. For two vectors $\mathbf{v}_1$ and $\mathbf{v}_2$, the inner product is defined as the dot product, i.e., $\langle \mathbf{v}_1, \mathbf{v}_2 \rangle := \mathbf{v}_1^\top \mathbf{v}_2$, and we use both notations interchangeably. For a matrix $\mathbf{M}$, its minimum and maximum eigenvalue are denoted by $\lambda_{\min}(\mathbf{M})$ and $\lambda_{\max}(\mathbf{M})$, respectively, and let $\mathrm{R}(\mathbf{M})$ denote the row space of $\mathbf{M}$, i.e., a subspace spanned by the rows of $\mathbf{M}$.

### 3.2 PROBLEM FORMULATION

In this section, we outline our problem setting and introduce several key assumptions. The true feature vector $\mathbf{z}_a \in \mathbb{R}^{d_z}$, associated with each arm $a \in [K]$, determines the rewards. However, the agent can observe only a subset of its elements, with the remaining elements unobserved. Specifically,

$\mathbf{z}_a$ is defined as follows:

$$\mathbf{z}_a := \left[ x_a^{(1)}, \cdots, x_a^{(d)}, u_a^{(1)}, \cdots, u_a^{(d_u)} \right]^\top. \tag{1}$$

where $\mathbf{x}_a := [x_a^{(1)}, \cdots, x_a^{(d)}]^\top \in \mathbb{R}^d$ refers to the *observable* part; $\mathbf{u}_a := [u_a^{(1)}, \cdots, u_a^{(d_u)}]^\top \in \mathbb{R}^{d_u}$ represents the *latent* part, which remains *inaccessible* to the agent. For clarity, the observable components will henceforth be highlighted in blue, while the unobservable components will be shown in red. Note that the dimensions of the latent feature vector, $d_u = d_z - d$, and the true feature vector, $d_z$, are both *unknown* to the agent. Consequently, the agent is unaware of whether the features are partially observed, which introduces significant challenges in selecting appropriate strategies. We use the following assumption regarding the features:

**Assumption 1** (Fixed features). *The true reward-generating features remain fixed throughout the entire decision horizon $T$ for all arms $a \in [K]$.*

This assumption is standard in the linear bandits with model misspecification error (Ghosh et al., 2017; Lattimore et al., 2020), which is a special case of our partially observable feature setting. In particular, if latent feature $\mathbf{u}_a$ arbitrarily changes over time, the problem itself would become non-learnable, making the problem ill-posed. Hence, given this, assuming fixed features is natural and well justified. (See Appendix B for details).

The reward associated with each arm is defined as the dot product of the corresponding true features $\mathbf{z}_a$ and an unknown parameter $\boldsymbol{\theta}_\star \in \mathbb{R}^{d_z}$, given by $y_{a,t} = \langle \mathbf{z}_a, \boldsymbol{\theta}_\star \rangle + \epsilon_t = \mathbf{z}_a^\top \boldsymbol{\theta}_\star + \epsilon_t$ for all $a \in [K]$. The error term, $\epsilon_t$, captures the inherent randomness in the reward, and we adopt a standard assumption commonly used in bandit problems for this error:

**Assumption 2** (Sub-Gaussian noise). *Let $\{\mathcal{F}_t\}_{t \in [T]}$ denote history at round $t$, represented by a filtration of sigma algebras. The reward noise $\epsilon_t$ is assumed to be a $\sigma$-sub-Gaussian random variable conditioned on $\mathcal{F}_t$. Formally, $\mathbb{E}[\exp(\lambda \epsilon_t) | \mathcal{F}_{t-1}] \leq \exp\left(\lambda^2 \sigma^2 / 2\right)$ for all $\lambda \in \mathbb{R}$.*

This assumption implies $\mathbb{E}[\epsilon_t | \mathcal{F}_{t-1}] = 0$, and $\mathbb{E}[y_{a,t} | \mathcal{F}_{t-1}] = \langle \mathbf{z}_a, \boldsymbol{\theta}_\star \rangle$. For brevity, we use $\mathbb{E}_{t-1}[\cdot]$ to denote $\mathbb{E}[\cdot | \mathcal{F}_{t-1}]$ henceforth. Given that $\epsilon_t$ is sampled after each action is observed, it follows that $\epsilon_t$ is $\mathcal{F}_t$-measurable. To eliminate issues of scale in the theoretical analysis, we assume that the expected reward $|\langle \mathbf{z}_a, \boldsymbol{\theta}_\star \rangle| \leq 1$ for all $a \in [K]$.

Let us write $\boldsymbol{\theta}_\star = [(\boldsymbol{\theta}_\star^{(o)})^\top, (\boldsymbol{\theta}_\star^{(u)})^\top]^\top$, where $\boldsymbol{\theta}_\star^{(o)} \in \mathbb{R}^d$ and $\boldsymbol{\theta}_\star^{(u)} \in \mathbb{R}^{d_u}$ are the parameters for observable features and latent features, respectively. Considering the composition of $\mathbf{z}_a$ defined in Eq. (1), we can decompose the reward $y_{a_t,t}$ into three terms as follows:

$$y_{a_t,t} = \langle \mathbf{x}_{a_t}, \boldsymbol{\theta}_\star^{(o)} \rangle + \epsilon_t + \langle \mathbf{u}_{a_t}, \boldsymbol{\theta}_\star^{(u)} \rangle \tag{2}$$

where the last term in Eq. (2) corresponds to the inaccessible portion of the reward. This reward model is equivalent to that imposed in the linear bandits with misspecification error (Lattimore et al., 2020). While the regret bound in Lattimore et al. (2020) includes misspecification error that grows linearly in decision horizon, our proposed method (Section 4) addresses this misspecification error and achieves a regret bound that is sublinear in the decision horizon.

Let $a_\star := \arg\max_{a \in [K]} \langle \mathbf{z}_a, \boldsymbol{\theta}_\star \rangle$ denote the optimal action, considering both observable and latent features. The theoretical performance of our algorithm is evaluated through cumulative regret, which measures the total expected difference between the reward of the optimal action and the reward of the action selected in each round. Formally,

$$\text{Reg}(T) = \mathbb{E}\left[ \sum_{t=1}^{T} (y_{a_\star,t} - y_{a_t,t}) \right] = \mathbb{E}\left[ \sum_{t=1}^{T} \langle \mathbf{x}_{a_\star} - \mathbf{x}_{a_t}, \boldsymbol{\theta}_\star^{(o)} \rangle + \sum_{t=1}^{T} \langle \mathbf{u}_{a_\star} - \mathbf{u}_{a_t}, \boldsymbol{\theta}_\star^{(u)} \rangle \right]. \tag{3}$$

Before introducing our method and algorithm, we first present a regret lower bound for a scenario where the inaccessible portion of the reward is ignored. For each $t \in [T]$, let $\pi_t$ denote policy that maps $\{\mathbf{x}_a : a \in [K]\}$ and $\{y_{a_s,s} : s \in [t-1]\}$ to a probability distribution over $[K]$. Then the policy is *dependent on the observed features* if there exist two sets of observed features $\{\mathbf{x}_a^{(1)} : a \in [K]\}$ and $\{\mathbf{x}_a^{(2)} : a \in [K]\}$ in $\mathbb{R}^{d \times K}$, such that for each given value of rewards $y_1, \ldots, y_{t-1} \in \mathbb{R}$, the policy is variant over the features, i.e.,

$$\pi_t(\mathbf{x}_1^{(1)}, \ldots, \mathbf{x}_K^{(1)}, y_1, \ldots, y_{t-1}) \neq \pi_t(\mathbf{x}_1^{(2)}, \ldots, \mathbf{x}_K^{(2)}, y_1, \ldots, y_{t-1}).$$

For instance, the UCB policy for linear bandits (with observed features) is dependent on the observed features, while the policy in the MAB algorithms (that disregard observed features) is not dependent on the observed features. In the theorem below, we particularly provide a lower bound for algorithms that employ policies that are dependent on the observed features.

**Theorem 1** (Regret lower bound ignoring latent reward component). *Suppose that $T \geq 4d^2$. For any algorithm $\Pi := (\pi_1, \ldots, \pi_T)$ that consists of policies $\{\pi_t : t \in [T]\}$ that are dependent on observed features, there exists a set of features $\{\mathbf{z}_1, \ldots, \mathbf{z}_K\}$ and a parameter $\boldsymbol{\theta}_\star \in \mathbb{R}^{d_z}$ such that the cumulative regret*

$$\text{Reg}_\Pi(T, \theta_\star, \mathbf{z}_1, \ldots, \mathbf{z}_K) \geq \frac{T}{3}.$$

This theorem implies that neglecting the latent portion of the reward in decision-making could result in regret that scales linearly with $T$, indicating a failure in the learning process of the agent. The comprehensive proof for this theorem is deferred to Appendix C.1.

## 4    ROBUST ESTIMATION FOR PARTIALLY OBSERVABLE FEATURES

We propose our estimation method to obtain a sublinear regret bound for linear bandits with latent features. Section 4.1 introduces the feature vector augmentation to handle the misspecification error and Section 4.2 presents the doubly robust estimation to further improve the regret bound.

### 4.1    FEATURE VECTOR AUGMENTATION WITH ORTHOGONAL PROJECTION

In order to minimize regret, it is sufficient to estimate the $K$ expected rewards $\{\mathbf{z}_a^\top \boldsymbol{\theta}_\star : a \in [K]\}$ rather than all components of $\boldsymbol{\theta}_\star \in \mathbb{R}^{d_z}$. A straightforward approach to this problem, which achieves a regret bound of $\widetilde{O}(\sqrt{KT})$, is to disregard the observed features and apply MAB algorithms like UCB1 (Auer et al., 2002). However, these algorithms tend to incur higher regret compared to those that leverage features, particularly when the number of arms is significantly larger than the dimension of the feature vectors, i.e., $K \gg d$.

We propose a unified approach to handle all cases of partially observable features and efficiently estimate all $K$ expected rewards. Let $\mathbf{X} := (\mathbf{x}_1, \ldots, \mathbf{x}_K) \in \mathbb{R}^{d \times K}$ represent a matrix that concatenates the observed part of the true features, and $\mathbf{U} := (\mathbf{u}_1^{(u)}, \ldots, \mathbf{u}_K^{(u)}) \in \mathbb{R}^{d_u \times K}$ represent the matrix that concatenates the latent complements of the true features for each arm. Without loss of generality, we assume a set of $K$ vectors $\{\mathbf{x}_1, \ldots, \mathbf{x}_K\}$ spans $\mathbb{R}^d$.[1] We define $\mathbf{P_X} := \mathbf{X}^\top (\mathbf{X}\mathbf{X}^\top)^{-1}\mathbf{X}$ as the projection matrix onto the row space of $\mathbf{X}$, denoted $\text{R}(\mathbf{X})$. Then the vector of rewards for all arms, $\mathbf{Y}_t = (y_{1,t}, \ldots, y_{K,t})$, is now decomposed as:

$$\begin{aligned}
\mathbf{Y}_t &= (\mathbf{X}^\top \boldsymbol{\theta}_\star^{(o)} + \mathbf{U}^\top \boldsymbol{\theta}_\star^{(u)}) + \epsilon_t \mathbf{1}_K \\
&= \mathbf{P_X}(\mathbf{X}^\top \boldsymbol{\theta}_\star^{(o)} + \mathbf{U}^\top \boldsymbol{\theta}_\star^{(u)}) + (\mathbf{I}_K - \mathbf{P_X})(\mathbf{X}^\top \boldsymbol{\theta}_\star^{(o)} + \mathbf{U}^\top \boldsymbol{\theta}_\star^{(u)}) + \epsilon_t \mathbf{1}_K \quad (4)\\
&= \mathbf{X}^\top (\boldsymbol{\theta}_\star^{(o)} + (\mathbf{X}\mathbf{X}^\top)^{-1}\mathbf{X}\mathbf{U}^\top \boldsymbol{\theta}_\star^{(u)}) + (\mathbf{I}_K - \mathbf{P_X})\mathbf{U}^\top \boldsymbol{\theta}_\star^{(u)} + \epsilon_t \mathbf{1}_K,
\end{aligned}$$

where the first and the second term are the projected rewards onto $\text{R}(\mathbf{X})$ and $\text{R}(\mathbf{X})^\perp$, the subspace of $\mathbb{R}^K$ perpendicular to $\text{R}(\mathbf{X})$. We write the projected parameter as $\boldsymbol{\mu}_\star^{(o)} := \boldsymbol{\theta}_\star^{(o)} + (\mathbf{X}\mathbf{X}^\top)^{-1}\mathbf{X}\mathbf{U}^\top \boldsymbol{\theta}_\star^{(u)}$.

Now we handle the second term in Eq. (4). For any set of basis $\{\mathbf{b}_1, \ldots, \mathbf{b}_{K-d}\} \in \text{R}(\mathbf{X})^\perp$, there exist $\mu_{\star,1}^{(u)}, \ldots, \mu_{\star,K-d}^{(u)}$ in $\mathbb{R}$ that express the projection of the reward as:

$$(\mathbf{I}_K - \mathbf{P_X})\mathbf{U}^\top \boldsymbol{\theta}_\star^{(u)} = \sum_{i=1}^{K-d} \mu_{\star,i}^{(u)} \mathbf{b}_i. \quad (5)$$

While the exact projected vector $(\mathbf{I}_K - \mathbf{P_X})\mathbf{U}^\top \boldsymbol{\theta}_\star^{(u)}$ is unknown, it is evident that the vector lies in the row space of $(\mathbf{I}_K - \mathbf{P_X})\mathbf{U}^\top$, whose dimension is:

$$d_h := \dim\left\{\text{R}\left((\mathbf{I}_K - \mathbf{P_X})\mathbf{U}^\top\right)\right\} = \dim(\text{R}(\mathbf{X})^\perp \cap \text{R}(\mathbf{U})) = \text{rank}(\mathbf{U}) - \dim(\text{R}(\mathbf{X}) \cap \text{R}(\mathbf{U})). \quad (6)$$

---

[1]When $d > K$, we can apply singular value decomposition on $\mathbf{X}$ to reduce the feature dimension to $\bar{d} \leq K$ with $\text{R}(\mathbf{X}) = \bar{d}$.

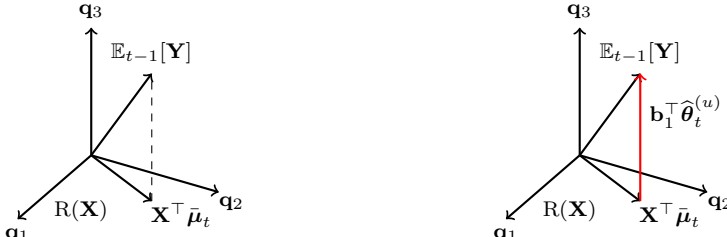

Figure 1: Illustration of the difference between conventional linear bandit algorithms (left) and our approach (right) in estimating rewards of $K = 3$ arms. The conventional algorithms use only observable features and find estimates on $\mathrm{R}(\mathbf{X})$ and the error due to unobserved features is accumulated. In contrast, our strategy projects the latent part of the reward onto the orthogonal complement of $\mathrm{R}(\mathbf{X})$ whose basis is denoted by $\mathbf{b}_1^\top \widehat{\boldsymbol{\theta}}_t^{(u)}$, and estimates the rewards of all arms in $\mathbb{R}^K$. Note that $\bar{\boldsymbol{\mu}}_t$ is the estimator of the parameter for observable features $\boldsymbol{\mu}_\star$.

Although the coefficients $\mu_{\star,1}^{(u)}, \ldots, \mu_{\star,K-d}^{(u)}$ depend on the choice of the basis vectors $\{\mathbf{b}_1, \ldots, \mathbf{b}_{K-d}\}$, at most $d_h$ coefficients are nonzero for any choice of the basis vectors. If we define $\boldsymbol{\mu}_\star$ as $[(\boldsymbol{\mu}_\star^{(o)})^\top, (\boldsymbol{\mu}_\star^{(u)})^\top]^\top \in \mathbb{R}^K$, where $\boldsymbol{\mu}_\star^{(u)} = [\mu_{\star,1}^{(u)}, \ldots, \mu_{\star,K-d}^{(u)}]^\top$, then Eq. (4) becomes $\mathbf{Y}_t = [\mathbf{X}^\top \; \mathbf{b}_1 \cdots \mathbf{b}_{K-d}]\boldsymbol{\mu}_\star + \epsilon_t \mathbf{1}_K$, implying that the reward for each $a \in [K]$ is:

$$y_{a,t} = \mathbf{e}_a^\top \mathbf{Y} = \mathbf{e}_a^\top [\mathbf{X}^\top \; \mathbf{b}_1 \cdots \mathbf{b}_{K-d}]\boldsymbol{\mu}_\star + \epsilon_t = [\mathbf{x}_a^\top \; \mathbf{e}_a^\top \mathbf{b}_1 \cdots \mathbf{e}_a^\top \mathbf{b}_{K-d}]\boldsymbol{\mu}_\star + \epsilon_t, \qquad (7)$$

where $\mathbf{e}_a \in \mathbb{R}^K$ is a standard basis, with elements all zero except for 1 in the $a$-th coordinate. With this modification, the rewards are now represented as a linear function of the augmented feature vectors, $\widetilde{\mathbf{x}}_a := [\mathbf{x}_a^\top \; \mathbf{e}_a^\top \mathbf{b}_1 \cdots \mathbf{e}_a^\top \mathbf{b}_{K-d}]^\top \in \mathbb{R}^K$, *without any misspecification error*. A toy example illustrating our strategy is shown in Figure 1.

The dimension of the augmented feature vectors $\{\widetilde{\mathbf{x}}_a : a \in [K]\}$ is $K \geq d$ and we propose an algorithm that employs the doubly robust ridge estimator and achieves $\widetilde{O}(\sqrt{KT})$ regret bound (see Appendix A). However, when $K > d$ and $d_u = 0$, the regret is high compared to the linear bandits with conventional features. Therefore, we propose a novel estimation strategy to avoid dependency on $K$ in the following section.

## 4.2 DOUBLY ROBUST LASSO ESTIMATOR

In Eq. (7), the parameter $\boldsymbol{\mu}_\star$ is sparse depending on the dimension of the latent features. Recall that $\boldsymbol{\mu}_\star^{(u)}$ are the coefficients to express the projection of the reward as represented in Eq. (5) and only $d_h$ basis vectors are required to express the projection of the reward; there are at most $d_h$ nonzero entries in $\boldsymbol{\mu}_\star^{(u)}$. The dimension $d_h$ reflects how closely the latent features are related to the observed features. Specifically, $d_h \leq \mathrm{rank}(\mathbf{U})$ where equality holds if and only if $\mathrm{R}(\mathbf{U}) \subseteq \mathrm{R}(\mathbf{X})^\perp$. Since $\mathrm{rank}(\mathbf{U}) \leq \min\{d_u, K\}$, the dimension $d_h$ cannot exceed $\min\{d_u, K\} - d$. Additionally, if $\mathrm{R}(\mathbf{U}) \subset \mathrm{R}(\mathbf{X})$, then $d_h = 0$.

Let $\check{\boldsymbol{\mu}}_t^L$ denote the Lasso estimator for $\boldsymbol{\mu}_\star$ using augmented feature vectors:

$$\check{\boldsymbol{\mu}}_t^L := \operatorname*{argmin}_{\boldsymbol{\mu}} \sum_{\tau=1}^t (y_{a_\tau} - \widetilde{\boldsymbol{x}}_{a_\tau}^\top \boldsymbol{\mu})^2 + 2\sigma \sqrt{\frac{2t}{p} \log \frac{2Kt^2}{\delta}} \left\| (\sum_{a \in [K]} \widetilde{\mathbf{x}}_a \widetilde{\mathbf{x}}_a^\top)^{1/2} \boldsymbol{\mu} \right\|_1. \qquad (8)$$

To enable the estimator in Eq. (8) to accurately detect the zero entries in $\boldsymbol{\mu}_\star$, the compatibility condition must be satisfied (van de Geer & Bühlmann, 2009). This condition holds when the minimum eigenvalue of the Gram matrix, $\lambda_{\min}\left(t^{-1} \sum_{s=1}^t \widetilde{\mathbf{x}}_{a_s} \widetilde{\mathbf{x}}_{a_s}^\top\right)$, is greater than zero. However, ensuring a sufficiently large minimum eigenvalue typically requires collecting a large number of exploration samples, which increases regret. Achieving this with fewer exploration samples is a critical challenge in bandit literature, as the minimum eigenvalue affects the convergence rate of the estimator and, consequently, the regret bound (Kim et al., 2021; Soare et al., 2014).

We introduce a doubly robust (DR) estimator that employs the *full feature* Gram matrix $\sum_{s=1}^{t} \sum_{a=1}^{K} \widetilde{\mathbf{x}}_a \widetilde{\mathbf{x}}_a^\top$ instead of $\sum_{s=1}^{t} \widetilde{\mathbf{x}}_{a_s} \widetilde{\mathbf{x}}_{a_s}^\top$. The DR estimation originates from the statistical literature on missing data, where "doubly robust" means that the estimator is robust against errors in the estimation of both the observation probability and the response model. In bandits, at each decision round $t \in [T]$, only the reward of the selected arm is observed, while the $K-1$ unselected rewards are *missing*. Thus DR estimation is applied to impute these $K-1$ missing rewards and include corresponding $K-1$ feature vectors in the estimation. Since the observation probability is given by the policy (which is known to the learner), the DR estimator is robust against errors in the estimated rewards. While Kim & Paik (2019) proposed a DR Lasso estimator on IID features that satisfies the compatibility condition, we propose another DR Lasso estimator that does not require the assumptions on the features.

We improve the DR estimation by incorporating resampling and coupling methods. In round $t$, the algorithm selects an action $a_t$ according to an $\epsilon_t$-greedy policy. Then, we generate a *pseudo-action* $\widetilde{a}_t$ from a multinomial distribution:

$$\phi_{a_t,t} := \mathbb{P}(\widetilde{a}_t = a_t | a_t) = p \quad \text{and} \quad \phi_{k,t} := \mathbb{P}(\widetilde{a}_t = k | a_t) = \frac{1-p}{K-1}, \ \forall k \in [K] \setminus \{a_t\}, \quad (9)$$

where $p \in (1/2, 1)$ is coupling probability set by the algorithm. To couple the policy of the actual action $a_t$ and the pseudo-action $\widetilde{a}_t$, we resample both of them until they match. This coupling yields a lower bound for the observation probability which reduces the variance of the DR pseudo-rewards in Eq. (10). Let $\mathcal{M}_t$ denote the event where $\widetilde{a}_t = a_t$ within a specified number of resamples. For given $\delta' \in (0, 1)$, we set the number of resamples as $\rho_t := \log((t+1)^2/\delta')/\log(1/p)$ so that event $\mathcal{M}_t$ occurs with probability at least $1 - \delta'/(t+1)^2$. Resampling allows the algorithm to explore further to find an action that balances between regret minimization and reward estimation.

This coupling replaces $\epsilon_t$ greedy policy with a multinomial distribution $\phi_{1,t}, \ldots, \phi_{K,t}$. When we use DR estimation with $\epsilon_t$ greedy policy, the inverse probability $\epsilon_t^{-1} := \sqrt{t}$ appears in the pseudo-reward (10), and thus the variance of the pseudo-reward explodes. Therefore, we couple the $\epsilon_t$ greedy policy with the multinomial distribution (9) to bound the inverse probability weight $\phi_{a,t}^{-1} = O(K)$.

With the pseudo-actions (coupled with the actual actions), we construct the unbiased pseudo-rewards for all arms $a \in [K]$,

$$\widetilde{r}_{a,t} := \widetilde{\mathbf{x}}_a^\top \check{\boldsymbol{\mu}}_t^L + \frac{\mathbb{I}(\widetilde{a}_t = a)}{\phi_{a,t}}\left(y_{a,t} - \widetilde{\mathbf{x}}_a^\top \check{\boldsymbol{\mu}}_t^L\right), \quad (10)$$

and note that $\check{\boldsymbol{\mu}}_t^L$ defined in Eq. (8) serves as the imputation estimator that fills in the missing rewards of unselected arms in round $t$.

For $a \neq \widetilde{a}_t$, i.e., an arm $a$ that is *not* selected in the round $t$, we impute the missing rewards using $\widetilde{\mathbf{x}}_k^\top \boldsymbol{\mu}_t^L$. For $a = \widetilde{a}_t$, however, the term $\mathbb{I}(\widetilde{a}_t = a)y_{a,t}/\phi_{a,t}$ calibrates the predicted reward to ensure the unbiasedness of the pseudo-rewards for all arms. Given that $\mathbb{E}_{\widetilde{a}_t}[\mathbb{I}(\widetilde{a}_t = a)] = \mathbb{P}(\widetilde{a}_t = a) = \phi_{a,t}$, taking the expectation over $\widetilde{a}_t$ on both sides of Eq. (10) gives $\mathbb{E}_{\widetilde{a}_t}[\widetilde{Y}_{a,t}] = \mathbb{E}_{t-1}[y_{a,t}] = \widetilde{\mathbf{x}}_a^\top \boldsymbol{\mu}_\star$ for all $a \in [K]$. Although the estimate $\widetilde{\mathbf{x}}_a^\top \check{\boldsymbol{\mu}}_t$ may have high error, it is multiplied by the mean-zero random variable $(1 - \mathbb{I}(\widetilde{a}_t = a)/\phi_{a,t})$, making the pseudo-rewards robust to the error in $\widetilde{\mathbf{x}}_a^\top \check{\boldsymbol{\mu}}_t$.

The pseudo-rewards can only be computed when $\widetilde{a}_t = a_t$, so they are used in rounds when the chosen action $a_t$ and the pseudo-action $\widetilde{a}_t$ match, indicated by the event $\mathcal{M}_t$. Since $\mathcal{M}_t$ occurs with high probability, we can compute the pseudo-rewards for almost all rounds. Our DR Lasso estimator is defined as:

$$\widehat{\boldsymbol{\mu}}_t^L := \underset{\boldsymbol{\mu}}{\text{argmin}} \sum_{\tau=1}^{t} \mathbb{I}(\mathcal{M}_\tau) \sum_{a \in [K]} \left(\widetilde{r}_{a,\tau} - \widetilde{\mathbf{x}}_a^\top \boldsymbol{\mu}\right)^2 + \frac{10\sigma}{p}\sigma\sqrt{2t\log\frac{2Kt^2}{\delta}}\left\|\left(\sum_{a \in [K]} \widetilde{\mathbf{x}}_a \widetilde{\mathbf{x}}_a^\top\right)^{1/2}\boldsymbol{\mu}\right\|_1,$$
$$(11)$$

and the following theorem provides a theoretical guarantee that this estimator converges across all arms after several exploration rounds.

**Theorem 2** (Consistency of the DR Lasso estimator)**.** *Let $d_h$ denote the dimension of the projected latent rewards defined in Eq. (6). For each $t$, let $\mathcal{E}_t \subseteq [t]$ denote an exploration phase such that for $\tau \in \mathcal{E}_t$ the action $a_\tau$ is sampled uniformly over $[K]$. Then for all round $t$ such that*

---

**Algorithm 1** Robust to Latent Feature (RoLF)

---

1: **INPUT:** features $\{\mathbf{x}_a : a \in [K]\}$, coupling probability $p \in (1/2, 1)$, confidence parameter $\delta > 0$.
2: Initialize $\widehat{\boldsymbol{\mu}}_0 = \mathbf{0}_K$, the exploration phase $\mathcal{E}_t = \emptyset$ and the exploration factor $C_{\mathrm{e}} := 8(\sqrt{K} + p^{-1})^2 p^2 (1-p)^{-2} K^2 \log \frac{2Kt^2}{\delta}$.
3: Find orthogonal basis $\mathbf{b}_1, \ldots, \mathbf{b}_{K-d}$ in $\mathrm{R}(\mathbf{X})^\perp$ to construct $\{\widetilde{\mathbf{x}}_a : a \in [K]\}$
4: **for** $t = 1, \ldots, T$ **do**
5:  **if** $|\mathcal{E}_t| \leq C_{\mathrm{e}} \log(2Kt^2/\delta)$ **then**
6:    Randomly sample $a_t$ uniformly over $[K]$ and $\mathcal{E}_t = \mathcal{E}_{t-1} \cup \{t\}$.
7:  **else**
8:    Compute $\widehat{a}_t := \arg\max_{a \in [K]} \widetilde{\mathbf{x}}_a^\top \widehat{\boldsymbol{\mu}}_{t-1}^L$
9:    **while** $\widetilde{a}_t \neq a_t$ and count $\leq \rho_t$ **do**
10:      Sample $a_t$ with $\mathbb{P}(a_t = \widehat{a}_t) = 1 - (t^{-1/2})$ and $\mathbb{P}(a_t = k) = t^{-1/2}/(K-1)$, $\forall k \neq \widehat{a}_t$.
11:      Sample $\widetilde{a}_t$ according to Eq. (9).
12:      count = count + 1
13:  Play $a_t$ and observe $y_{a_t, t}$.
14:  **if** $\widetilde{a}_t \neq a_t$ **then**
15:    Set $\widehat{\boldsymbol{\mu}}_t^L := \widehat{\boldsymbol{\mu}}_{t-1}^L$
16:  **else**
17:    Update $\widehat{\boldsymbol{\mu}}_t^L$ following Eq. (11) with $\widetilde{r}_{a,t}$ and update $\check{\boldsymbol{\mu}}_t^L$ following Eq. (8).

---

$t \geq |\mathcal{E}_t| \geq 8(\sqrt{K} + p^{-1})^2 p^2 (1-p)^{-2} K^2 \log \frac{2Kt^2}{\delta}$, *with probability at least* $1 - 2\delta/t^2$,

$$\max_{a \in [K]} |\widetilde{\mathbf{x}}_a(\widehat{\boldsymbol{\mu}}_t - \boldsymbol{\mu}_\star)| \leq \frac{20\sigma}{p} \sqrt{\frac{2(d + d_h) \log \frac{2Kt^2}{\delta}}{t}}, \tag{12}$$

Although we use $K$-dimensional feature vectors, the error bound of the DR Lasso estimator is only logarithmic in $K$. This fast convergence rate is possible with the regularity conditions, such as the restrictive minimum eigenvalue condition (Bühlmann & Van De Geer, 2011; van de Geer & Bühlmann, 2009). While conventional method assumes that the feature vectors satisfy the condition, our approach does not require this assumption, since our augmented features are orthogonal vectors in $\mathrm{R}(\mathbf{X})^\perp$, their average Gram matrix satisfies $\lambda_{\min}(\sum_{a \in [K]} \widetilde{\mathbf{x}}_a \widetilde{\mathbf{x}}_a^\top) \geq \min\{1, \lambda_{\min}(\sum_{a \in [K]} \mathbf{x}_a \mathbf{x}_a^\top)\}$. Thus, the convergence rate has only $\sqrt{\log K}$ rate in terms of $K$.

The consistency is proved by bounding the two components of the error in the pseudo-rewards defined in (10): (i) the noise of the reward and (ii) the error of the imputation estimator $\check{\mu}_t$. Since (i) is sub-Gaussian, it can be bounded using martingale inequalities. For (ii), the imputation error $\widetilde{x}_a^\top (\check{\mu}_t^L - \mu_\star)$ is multiplied by the mean-zero random variable $\left(1 - \frac{\mathbb{I}(\widetilde{a}_t = a)}{\phi_{a,t}}\right)$ and thus it can be bounded by $\|\check{\mu}_t^L - \mu_\star\|_1/\sqrt{t}$.

## 5 PROPOSED ALGORITHM AND THEORETICAL ANALYSIS

In this section, we present our proposed algorithm, which is based on a novel estimation approach for handling partially observable features. The proposed algorithm significantly improves the regret bound compared to linear bandit algorithms, which rely solely on observed features, and MAB algorithms, even without prior knowledge of the latent features.

### 5.1 ROBUST TO LATENT FEATURES (RoLF) ALGORITHM

In the initialization step, when the observable features are given, our algorithm finds a set of orthogonal basis $\{\mathbf{b}_1, \ldots, \mathbf{b}_{K-d}\} \in \mathrm{R}(\mathbf{X})^\perp$ to augment each observable features. After the exploration phase, the algorithm computes the candidate action, denoted by $\widehat{a}_t$, and then resample both $\widetilde{a}_t$ and $a_t$ until they match. If $\widetilde{a}_t$ and $a_t$ do not match within $\rho_t := \log((t+1)^2/\delta')/\log(1/p)$, the resampling phase ends, and the agent selects $a_t$ and observes $y_{a_t, t}$. If they match, the algorithm updates both the imputation and main estimators according to the equations provided in Eq. (8) and Eq. (11).

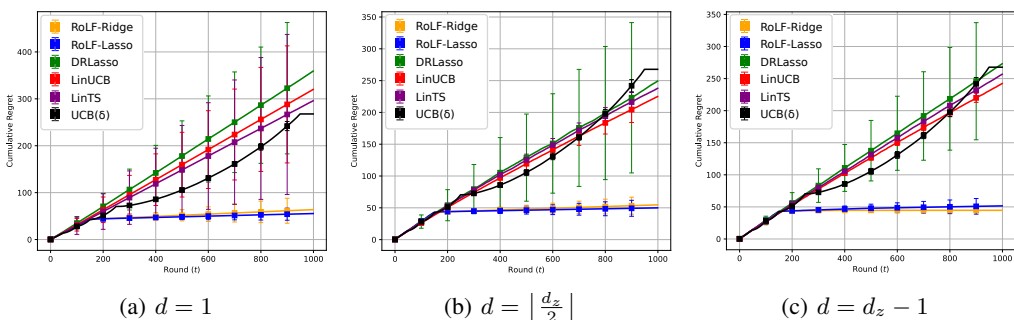

(a) $d = 1$        (b) $d = \left\lfloor \frac{d_z}{2} \right\rfloor$        (c) $d = d_z - 1$

Figure 2: Cumulative regrets of the algorithms in comparison for scenario 1 ($K = 50, d_z = 31$).

The proposed algorithm does not require the knowledge of the dimension of the latent features $d_u$ and the dimension of the projected rewards from latent feature space onto the $\mathrm{R}(\mathbf{X})^\perp$. Although we present the algorithm on fixed feature vectors, the algorithm applies to arbitrary feature vectors that changes over time by updating the orthogonal basis.

## 5.2 REGRET ANALYSIS

We provide an analysis of `RoLF` using the Lasso estimators, as detailed in the following theorem:

**Theorem 3** (Regret bound for Lasso `RoLF`). *Let $d_h$ denote the dimension of the projected latent rewards defined in Eq. (6). Then for $\delta \in (0, 1)$ and $p \in (1/2, 1)$, the expected cumulative regret of the proposed algorithm is bounded by*

$$
\begin{aligned}
\mathbb{E}[\mathrm{Reg}(T)] \leq\; & 10\delta \log T + \frac{16K^2(\sqrt{K} + p^{-1})^2}{(1-p)^2} + \frac{2p\sqrt{T}}{K-1} \frac{\log \frac{(T+1)^2}{\delta}}{\log(1/p)} \\
& + \frac{80\sigma}{p} \sqrt{2(d + d_h)T \log \frac{2KT^2}{\delta}},
\end{aligned}
\tag{13}
$$

To the best of our knowledge, Theorem 3 is the first regret bound sublinear in $T$ for the latent features without any structural assumption. With slight modifications, the regret bound can also be applied to scenarios with time-varying features and misspecified linear bandit problems.

Note that the number of rounds for the exploration phase is $O(K^3 \log KT)$, which is only logarithmic in the horizon $T$. The factor $K^3$ is not reducible since the algorithm must estimate all $K$ biases from the missing features. Using the Gram matrix with *full feature vectors*, $\sum_{a=1}^{K} \widetilde{\mathbf{x}}_a \widetilde{\mathbf{x}}_a^\top$ in combination with DR estimation reduces the exploration phase time from $O(K^4 \log KT)$ to $O(K^3 \log KT)$, reducing the complexity by a factor of $K$. The convergence rate in the last term is proportional to $\sqrt{d + d_h}$ rather than $\sqrt{K}$, as shown in Eq. (12). Thus, our regret bound is $O(\sqrt{(d + d_h)T \log KT})$.

## 6 NUMERICAL EXPERIMENTS

In this experiment, we simulate and compare two versions of our algorithm, presented in Algorithm 1 and Algorithm 2 (Appendix A), with linear bandit algorithms that use only observable features: `LinUCB` (Li et al., 2010; Chu et al., 2011) and `LinTS` (Agrawal & Goyal, 2013). These algorithms use the UCB and Thompson sampling methods, respectively, when the reward is modeled as a linear function of the features. Additionally, since our algorithm incorporates DR estimation with the Lasso estimator, we include `DRLasso` (Kim & Paik, 2019) in the comparison as well. To further evaluate the performance of our algorithm in scenarios where latent features are expected but ignored, we also compare it with `UCB`($\delta$) (Lattimore & Szepesvári, 2020), an MAB algorithm without features.

For the simulation environment, we generate true features $\mathbf{z}_a$ for each arm $a \in [K]$ from $\mathcal{N}(\mathbf{0}, \mathbf{I}_{d_z})$ and subsample $d$ elements to construct $\mathbf{x}_a$. Orthogonal basis vectors $\{\mathbf{b}_1, \ldots, \mathbf{b}_{K-d}\}$ are derived via singular value decomposition (SVD) on the observable feature matrix $\mathbf{X}$, ensuring orthogonality

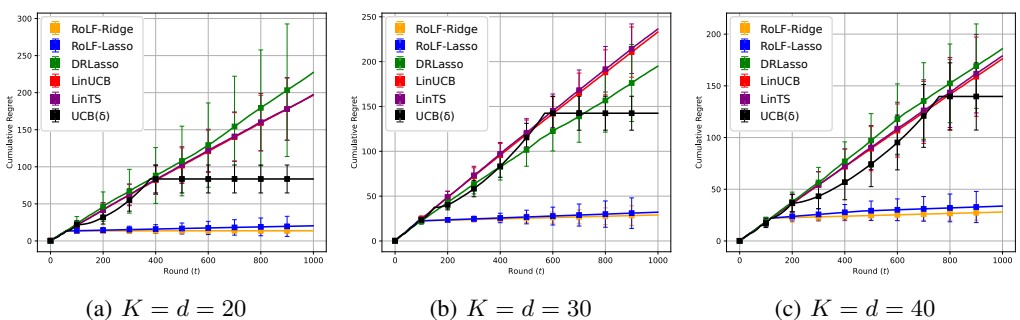

(a) $K = d = 20$      (b) $K = d = 30$      (c) $K = d = 40$

Figure 3: Cumulative regrets of the algorithms in comparison for scenario 2 ($d_z = 60$).

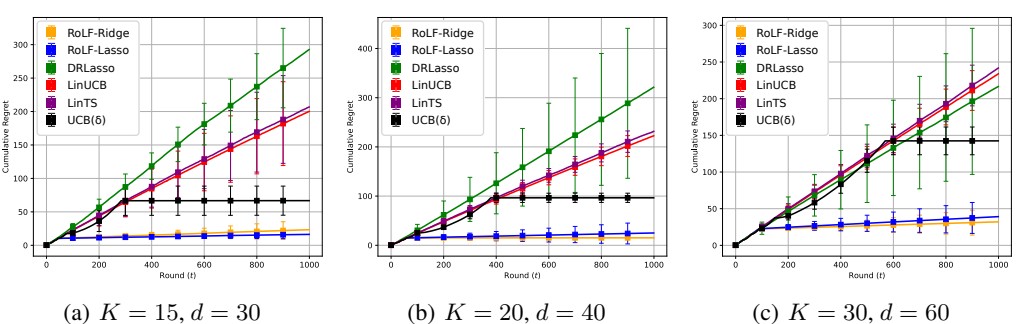

(a) $K = 15, d = 30$      (b) $K = 20, d = 40$      (c) $K = 30, d = 60$

Figure 4: Cumulative regrets of the algorithms in comparison for scenario 3 ($d_z = d$).

to $R(\mathbf{X})$. We augment $\mathbf{X}$ with the basis vectors via linear concatenation. Rewards are generated by sampling the unknown parameter $\boldsymbol{\theta}_\star \in \mathbb{R}^k$ from $\mathrm{Unif}(-1/2, 1/2)$. The hyperparameter $p$, for the sampling distribution of $\widetilde{a}_t$, is set to 0.6 (see Eq. (9)). The confidence parameter $\delta$ is 0.0001, and the total decision horizon is $T = 1000$. To address both partial and full observability, $d_z \geq d$ is used, and we run 5 independent experiments. We compare the algorithms across three scenarios:

**Scenario (i).** We examine algorithm performance as $d$, the number of observed elements, varies to assess the impact of observability. With $K = 50$ arms and $d_z = 31$, we compare results for $d = 1, \lfloor d_z/2 \rfloor = 15$, and $d_z - 1 = 30$. Figure 2 presents the results, showing that our algorithm consistently outperforms others in regret and robustness. In contrast, LinUCB, LinTS, and DRLasso show significant dependence on the number of observable features, with performance deteriorating and variability increasing as observability decreases.

**Scenario (ii).** In this scenario, the number of arms is equal to the dimension of the observed features, i.e., $K = d$. The main objective of this experiment is to demonstrate that our algorithm remains robust to changes in the number of arms $K$, unlike MAB algorithms that disregard the observable features. Specifically, we compare the algorithms with $K$ set to 20, 30, and 40, and $d_z = 60$ for all cases. Figure 3 shows the results for this scenario. As we can observe, the performance of UCB($\delta$) worsens as the number of arms increases in each environment. In contrast, our algorithm shows better performance in terms of both the level of regret and robustness.

**Scenario (iii).** We evaluate performance when the number of arms is less than the dimension of observed features, setting $d = 2K$ and varying $K$ as 15, 20, and 30, with $d_z = d$. Before using the features in our algorithms, we apply singular value decomposition (SVD) for dimensionality reduction. Figure 4 shows that our algorithm performs well even in extreme cases. By applying dimension reduction through SVD, our algorithm remains applicable even when the matrix of feature vectors is not full rank. Furthermore, the results suggest that our algorithm demonstrates superior performance even in the absence of partial observability.

REPRODUCIBILITY STATEMENT

All theoretical results made in this paper are accompanied by detailed proofs, which can be found in Appendix C and Appendix D. The assumptions underlying these claims are clearly stated in Section 3.2 of the main text. Furthermore, for the implementation of our proposed algorithm, along with instructions for reproducing the experimental results, we provide a ZIP file containing the source code in the supplementary materials.

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

---

**Algorithm 2** Robust to Latent Feature with Ridge Estimator (RoLF-Ridge)

---

1: **INPUT:** features $\{\mathbf{x}_a : a \in [K]\}$, coupling probability $p \in (1/2, 1)$, confidence parameter $\delta > 0$.
2: Initialize $\widehat{\boldsymbol{\mu}}_0 = \mathbf{0}_K$, the exploration phase $\mathcal{E}_t = \emptyset$ and the exploration factor $C_e := 32(1 - p)^{-2} K^2$.
3: Find orthogonal basis $\mathbf{b}_1, \ldots, \mathbf{b}_{K-d}$ in $\mathrm{R}(\mathbf{X})^{\perp}$ to construct $\{\widetilde{\mathbf{x}}_a : a \in [K]\}$
4: **for** $t = 1, \ldots, T$ **do**
5:     **if** $|\mathcal{E}_t| \leq C_e \log(2Kt^2/\delta)$ **then**
6:         Randomly sample $a_t$ uniformly over $[K]$ and $\mathcal{E}_t = \mathcal{E}_{t-1} \cup \{t\}$.
7:     **else**
8:         Compute $\widehat{a}_t := \arg\max_{a \in [K]} \widetilde{\mathbf{x}}_a^\top \widehat{\boldsymbol{\mu}}_{t-1}^R$
9:         **while** $\widetilde{a}_t \neq a_t$ and count $\leq \rho_t$ **do**
10:             Sample $a_t$ with $\mathbb{P}(a_t = \widehat{a}_t) = 1 - (t^{-1/2})$ and $\mathbb{P}(a_t = k) = t^{-1/2}/(K-1)$, $\forall k \neq \widehat{a}_t$.
11:             Sample $\widetilde{a}_t$ according to Eq. (9).
12:             count = count + 1
13:     Play $a_t$ and observe $y_{a_t, t}$.
14:     **if** $\widetilde{a}_t \neq a_t$ **then**
15:         Set $\widehat{\boldsymbol{\mu}}_t^R := \widehat{\boldsymbol{\mu}}_{t-1}^R$
16:     **else**
17:         Update $\widehat{\boldsymbol{\mu}}_t^R$ following Eq. (14) with $\widetilde{r}_{a,t}$ and update $\check{\boldsymbol{\mu}}_t^R$ following Eq. (15).

---

# A ROBUST TO LATENT FEATURE ALGORITHM WITH RIDGE ESTIMATOR

Our Doubly robust (DR) ridge estimator is defined as follows:

$$\widehat{\boldsymbol{\mu}}_t^R := \left( \sum_{\tau=1}^t \mathbb{I}(\mathcal{M}_\tau) \sum_{a \in [K]} \widetilde{\mathbf{x}}_a \widetilde{\mathbf{x}}_a^\top + I_K \right)^{-1} \left( \sum_{\tau=1}^t \mathbb{I}(\mathcal{M}_\tau) \sum_{a \in [K]} \widetilde{\mathbf{x}}_a \widetilde{r}_{a,\tau} \right), \tag{14}$$

where $\widetilde{r}_{a,\tau}$ is the DR pseudo reward:

$$\widetilde{r}_{a,t} := \widetilde{\mathbf{x}}_a^\top \check{\boldsymbol{\mu}}_t^R + \frac{\mathbb{I}(\widetilde{a}_t = a)}{\phi_{a,t}} \left( y_{a,t} - \widetilde{\mathbf{x}}_a^\top \check{\boldsymbol{\mu}}_t^R \right),$$

and the imputation estimator $\check{\boldsymbol{\mu}}_t^R$ is defined as

$$\check{\boldsymbol{\mu}}_t^R := \left( \sum_{\tau=1}^t \widetilde{\mathbf{x}}_{a_\tau} \widetilde{\mathbf{x}}_{a_\tau}^\top + p\mathbf{I}_K \right)^{-1} \left( \sum_{\tau=1}^t \widetilde{\mathbf{x}}_{a_\tau} y_{a_\tau, \tau} \right). \tag{15}$$

The following theorem shows that this Ridge estimator is consistent, meaning it converges to the true parameter $\boldsymbol{\mu}_\star$ with high probability as the agent interacts with the environment.

**Theorem 4** (Consistency of the DR Ridge estimator). *For each $t$, let $\mathcal{E}_t \subseteq [t]$ denote an exploration phase such that for $\tau \in \mathcal{E}_t$ the action $a_\tau$ is sampled uniformly over $[K]$. Then for all round $t$ such that $|\mathcal{E}_t| \geq 32(1-p)^{-2} K^2 \log(2Kt^2/\delta)$, with probability at least $1 - 3\delta$,*

$$\max_{a \in [K]} |\widetilde{\mathbf{x}}_a^\top (\widehat{\boldsymbol{\mu}}_t^R - \boldsymbol{\mu}_\star)| \leq \frac{2}{\sqrt{t}} \left( \frac{\sigma}{p} \sqrt{K \log \frac{t+1}{\delta}} + \sqrt{K} \right).$$

With $|\mathcal{E}_t| = O(K^2 \log Kt)$ number of exploration, the DR Ridge estimator achieves $O(\sqrt{K/t})$ convergence rate over all $K$ rewards. This is possible because the DR pseudo-rewards defined in Eq. (10) impute the missing rewards for all arms $a \in [K]$ using $\widetilde{\mathbf{x}}_a^\top \check{\boldsymbol{\mu}}_t$, based on the samples collected during the exploration phase, $\mathcal{E}_t$. With this convergence guarantee, we establish a regret bound for RoLF-Ridge, which is the adaptation of Algorithm 1 using the Ridge estimator.

**Theorem 5** (Regret bound for Ridge RoLF). *For $\delta \in (0, 1)$, the expected cumulative regret of the proposed algorithm using DR Ridge estimator is bounded by*

$$\mathrm{Reg}(T) \leq 6\delta \log T + \frac{2p\sqrt{T}}{K-1} \frac{\log \frac{(T+1)^2}{\delta}}{\log(1/p)} + \frac{32K^2}{(1-p)^2} \log \frac{2dT^2}{\delta} + 8\sqrt{KT} \left( \frac{\sigma}{p} \sqrt{\log \frac{T^2}{\delta}} + 1 \right).$$

The first and second terms come from the distribution of $a_t$ which is a combination of the $1 - t^{-1/2}$-greedy policy and resampling up to $\rho_t := \log((t+1)^2/\delta)/\log(1/p)$ trials. The third term is determined by the size of the exploration set, $\mathcal{E}_t$, while the last term arises from the estimation error bounded by the DR estimator as described in Theorem 4. The hyperparameter $p \in (1/2, 1)$ balances the size of the exploration set in the third term and the estimation error in the last term. Overall, the regret is $O(\sqrt{KT \log T})$, which shows a significant improvement compared to the regret lower bound in Theorem 1 for any linear bandit algorithms that do not account for unobserved features and unobserved rewards.

# B  A MODIFIED ALGORITHM FOR TIME-VARYING OBSERVED FEATURES

In this section, we propose an algorithm for linear bandits with partially observable features under the setting where the observed features vary over time.

## B.1  PROBLEM FORMULATION

Let $\mathbf{x}_{1,t}, \ldots, \mathbf{x}_{K,t}$ denote the observed features and $\mathbf{u}_1, \ldots, \mathbf{u}_K$ denote the unobservable features. Now the observed features arbitrary changes over $t$ but the unobservable features are fixed over time. When the algorithms selects an arm $a_t$, the reward is

$$y_{a_t,t} = \langle \mathbf{x}_{a_t,t}, \boldsymbol{\theta}_\star^{(o)} \rangle + \langle \mathbf{u}_{a_t}, \boldsymbol{\theta}_\star^{(u)} \rangle + \epsilon_t,$$

where the $\epsilon_t$ is the Sub-Gaussian noise that follows Assumption 2. Under Assumption 1, the features are fixed and the expected reward of each arm is also fixed over time, where MAB algorithms without using features are applicable to achieve $\widetilde{O}(\sqrt{KT})$ regret bound. When the observed features vary over time, the expected reward of each arm $\mathbb{E}[y_{a_t}, t] = \langle \mathbf{x}_{a_t,t}, \boldsymbol{\theta}_\star^{(o)} \rangle + \langle \mathbf{u}_{a_t}, \boldsymbol{\theta}_\star^{(u)} \rangle$ also arbitrarily changes over time and MAB algorithms suffer regret linear in $T$. To our knowledge, there is no other work that address this challenging setting.

## B.2  PROPOSED METHOD: ORTHOGONAL BASIS AUGMENTATION

We address the problem by augmenting Euclidean basis $\mathbf{e}_1, \ldots, \mathbf{e}_K$ in $RR^K$ to estimate bias caused by the unobservable features. Let $\widetilde{\mathbf{x}}_{a,t} := \mathbf{e}_a^\top [\mathbf{X}_t \; \mathbf{e}_1 \; \cdots \; \mathbf{e}_K] \in \mathbb{R}^{d+K}$ and let $\Delta_a := \langle \mathbf{u}_a, \theta_\star^{(u)} \rangle$ denote the bias stems from the latent features. Then,

$$
\begin{aligned}
y_{a,t} &= \langle \mathbf{x}_{a,t}^\top \boldsymbol{\theta}_\star^{(o)} \rangle + \langle \mathbf{u}_{a,t}^\top \boldsymbol{\theta}_\star^{(u)} \rangle + \epsilon_{a,t} \\
&= \langle \mathbf{e}_a^\top [\mathbf{X}_t \mathbf{e}_1 \cdots \mathbf{e}_K], [\boldsymbol{\theta}_\star^{(o)} \Delta_1 \cdots \Delta_K] \rangle + \epsilon_{a,t}.
\end{aligned}
$$

Therefore, applying the `RoLF-Ridge` algorithm to the new features $\widetilde{\mathbf{x}}_{a,t} := \mathbf{e}_a^\top [\mathbf{X}_t \mathbf{e}_1 \cdots \mathbf{e}_K]$ yields the following regret bound.

**Theorem 6** (Regret bound for `Ridge-RoLF-V` with time varying observed features). *If observed features are vary over time, for $\delta \in (0,1)$, the expected cumulative regret of the proposed algorithm* `Ridge-RoLF-V` *using DR Ridge estimator is bounded by*

$$
\begin{aligned}
\text{Reg}(T) \leq & 6\delta \log T + \frac{2p\sqrt{T}}{d+K-1} \frac{\log \frac{(T+1)^2}{\delta}}{\log(1/p)} + \frac{32(K+d)^2}{(1-p)^2} \log \frac{2(K+d)T^2}{\delta} \\
& + 8\sqrt{(d+K)T} \left( \frac{\sigma}{p} \sqrt{\log \frac{T^2}{\delta}} + 1 \right).
\end{aligned}
$$

The proof is similar to that in Theorem 5 and we omit the proof. The rate of the regret bound is $\widetilde{O}(\sqrt{(d+K)T})$ and, to our knowledge, this is the first sublinear regret bound for the partially observable linear bandits (as well as misspecified linear bandits) with arbitral time-varying observed features.

# C  MISSING PROOFS

## C.1  PROOF OF THEOREM 1

We start the proof by providing a detailed account of the scenario described in the theorem. Without loss of generality, we consider the case where $K = 3$. As stated in the theorem, $a_\star$ represents the index of the optimal action when considering the entire reward, including both observable and latent components. In contrast, $a_o$ denotes the index of the optimal action when considering only the observable component. For the sake of clarity in the proof, we introduce an additional notation, $a'$, which refers to an action whose observable features are identical to those of $a_\star$, but with a distinct latent component. Specifically, this implies that $a' \neq a_\star$ and $\mathbf{z}_{a'} \neq \mathbf{z}_{a_\star}$, but $\mathbf{x}_{a'} = \mathbf{x}_{a_\star}$.

Taking this scenario into account, the observable part of the features associated with $a_\star$, $a'$, and $a_o$ are defined as follows:

$$\mathbf{x}_{a_\star} := \left[ -\frac{d}{\sqrt{T}}, \ldots, -\frac{d}{\sqrt{T}} \right]^\top, \mathbf{x}_{a'} := \left[ -\frac{d}{\sqrt{T}}, \ldots, -\frac{d}{\sqrt{T}} \right]^\top, \mathbf{x}_{a_o} := \left[ \frac{d}{\sqrt{T}}, \ldots, \frac{d}{\sqrt{T}} \right]^\top.$$

Additionally, we assume that the unobservable portion of the true features, $\mathbf{u}_a \in \mathbb{R}^{d_u}$, is drawn from the set $\mathcal{U} := \{-1, 1\}^{d_u}$. We define the unobservable feature vectors for actions $a_\star$, $a'$, and $a_o$ as follows:

$$\mathbf{u}_{a_\star} := [1, \ldots, 1]^\top, \mathbf{u}_{a'} := [-1, \ldots, -1]^\top, \mathbf{u}_{a_o} := [-1, \ldots, -1]^\top,$$

where in $\mathbf{u}_{a'}$, the number of 1's and -1's are equal. Since $T \geq 4d^2$, it can be observed that the supremum norms of $\mathbf{z}_{a_\star}$, $\mathbf{z}_{a'}$, and $\mathbf{z}_{a_o}$ — each constructed by concatenating the observable and corresponding unobservable parts — do not exceed 1. This ensures that the scenario aligns with the assumption imposed on the feature vectors throughout this paper.

We further define the true parameter, incorporating the definition of $\boldsymbol{\theta}_\star^{(o)}$ from the theorem statement:

$$\boldsymbol{\theta}_\star := \left[ \frac{1}{3d}, \ldots, \frac{1}{3d}, \frac{2}{3d_u}, \ldots, \frac{2}{3d_u} \right]^\top.$$

Note that it is straightforward to verify that $\|\boldsymbol{\theta}_\star\|_1 = 1$, thereby satisfying $|\langle \mathbf{z}_a, \boldsymbol{\theta}_\star \rangle| \leq 1$. With this established, we can also observe that the expected reward for the three actions are defined as:

$$\langle \mathbf{z}_{a_\star}, \boldsymbol{\theta}_\star \rangle = \langle \mathbf{x}_{a_\star}, \boldsymbol{\theta}_\star^{(o)} \rangle + \langle \mathbf{u}_{a_\star}, \boldsymbol{\theta}_\star^{(u)} \rangle = -\frac{d}{3\sqrt{T}} + \frac{2}{3},$$

$$\langle \mathbf{z}_{a'}, \boldsymbol{\theta}_\star \rangle = \langle \mathbf{x}_{a'}, \boldsymbol{\theta}_\star^{(o)} \rangle + \langle \mathbf{u}_{a'}, \boldsymbol{\theta}_\star^{(u)} \rangle = -\frac{d}{3\sqrt{T}} - \frac{2}{3},$$

$$\langle \mathbf{z}_{a_o}, \boldsymbol{\theta}_\star \rangle = \langle \mathbf{x}_{a_o}, \boldsymbol{\theta}_\star^{(o)} \rangle + \langle \mathbf{u}_{a_o}, \boldsymbol{\theta}_\star^{(u)} \rangle = \frac{d}{3\sqrt{T}} - \frac{2}{3},$$

respectively. Given the assumption $T \geq 4d^2$, we can verify that:

$$\langle \mathbf{z}_{a_\star}, \boldsymbol{\theta}_\star \rangle - \langle \mathbf{z}_{a_o}, \boldsymbol{\theta}_\star \rangle = \frac{4}{3} - \frac{2d}{3\sqrt{T}} \geq 1, \tag{16}$$

which confirms that $a_\star$ is optimal when considering the full feature set.

Using the conventional linear bandit algorithms such as `OFUL` (Abbasi-yadkori et al., 2011) and `LinTS` (Agrawal & Goyal, 2013), the action selected in round $t$, $a_t$, is based solely on $\mathbf{x}_{a_t}$, thereby neglecting the unobserved portion of the reward. Given this action $a_t$, the instantaneous regret incurred by these algorithms in round $t$ is defined and decomposed as follows:

$$\begin{aligned} \text{Reg}(t) &= \langle \mathbf{z}_{a_\star}, \boldsymbol{\theta}_\star \rangle - \langle \mathbf{z}_{a_t}, \boldsymbol{\theta}_\star \rangle \\ &= \underbrace{\langle \mathbf{z}_{a_\star} - \mathbf{z}_{a_o}, \boldsymbol{\theta}_\star \rangle}_{(*)} + \langle \mathbf{z}_{a_o} - \mathbf{z}_{a_t}, \boldsymbol{\theta}_\star \rangle. \end{aligned}$$

We consider the first part denoted by $(*)$. Note that this term is calculated as described in Eq. (16), and is therefore lower bounded by 1. Hence, the cumulative regret becomes:

$$
\begin{aligned}
\mathrm{Reg}(T) &= \sum_{t=1}^{T} \mathrm{Reg}(t) \\
&= \sum_{t=1}^{T} \left( \langle \mathbf{z}_{a_\star} - \mathbf{z}_{a_o}, \boldsymbol{\theta}_\star \rangle + \langle \mathbf{z}_{a_o} - \mathbf{z}_{a_t}, \boldsymbol{\theta}_\star \rangle \right) \\
&\geq \sum_{t=1}^{T} 1 + \sum_{t=1}^{T} \langle \mathbf{z}_{a_o} - \mathbf{z}_{a_t}, \boldsymbol{\theta}_\star \rangle \\
&= T + \underbrace{\sum_{t=1}^{T} \langle \mathbf{z}_{a_o} - \mathbf{z}_{a_t}, \boldsymbol{\theta}_\star \rangle}_{(**)}.
\end{aligned}
\tag{17}
$$

For the term denoted by $(**)$, it can be further decomposed as follows:

$$
\sum_{t=1}^{T} \left( \langle \mathbf{x}_{a_o}, \boldsymbol{\theta}_\star^{(o)} \rangle - \langle \mathbf{x}_{a_t}, \boldsymbol{\theta}_\star^{(o)} \rangle \right) + \sum_{t=1}^{T} \left( \langle \mathbf{u}_{a_o}, \boldsymbol{\theta}_\star^{(u)} \rangle - \langle \mathbf{u}_{a_t}, \boldsymbol{\theta}_\star^{(u)} \rangle \right),
$$

where the first term corresponds to the regret induced by linear bandit algorithms that only consider observable features, and by definition, this term is always greater than or equal to 0.

The second term of this decomposition depends on how often $a_t$ matches $a_\star$, since selecting $a_\star$ makes this term negative. Following the definitions of $\mathbf{x}_{a_\star}$ and $\mathbf{x}_{a'}$, the agent cannot accurately distinguish between the two actions when their respective latent rewards are excluded. As a result, one of the two actions is chosen uniformly at random, meaning $a_\star$ is selected at most $T/2$ times in the worst-case scenario. Thus, the second term is bounded below by $-2T/3$, leading to the following inequality:

$$
\begin{aligned}
T + \sum_{t=1}^{T} \langle \mathbf{z}_{a_o} - \mathbf{z}_{a_t}, \boldsymbol{\theta}_\star \rangle &\geq T + \sum_{t=1}^{T} \left( \langle \mathbf{x}_{a_o}, \boldsymbol{\theta}_\star^{(o)} \rangle - \langle \mathbf{x}_{a_t}, \boldsymbol{\theta}_\star^{(o)} \rangle \right) - \frac{2}{3} T \\
&\geq \frac{T}{3} + \sum_{t=1}^{T} \left( \langle \mathbf{x}_{a_o}, \boldsymbol{\theta}_\star^{(o)} \rangle - \langle \mathbf{x}_{a_t}, \boldsymbol{\theta}_\star^{(o)} \rangle \right) \\
&\geq \frac{T}{3},
\end{aligned}
$$

which completes the proof. $\qquad\square$

## C.2   PROOF OF THEOREM 2

Let $\mathbf{V}_t := \sum_{\tau=1}^{t} \sum_{a \in [K]} \widetilde{\mathbf{x}}_a \widetilde{\mathbf{x}}_a^\top$. Then

$$
\max_{a \in [K]} |\widetilde{\mathbf{x}}_a^\top (\widehat{\boldsymbol{\mu}}_t - \boldsymbol{\mu}_\star)| \leq \sqrt{\sum_{a \in [K]} |\widetilde{\mathbf{x}}_a^\top (\widehat{\boldsymbol{\mu}}_t - \boldsymbol{\mu}_\star)|^2} = t^{-1/2} \|\widehat{\boldsymbol{\mu}}_t - \boldsymbol{\mu}_\star\|_{\mathbf{V}_t}
$$

Recall that $\widehat{\mathbf{w}}_t := \mathbf{V}_t^{1/2} \widehat{\boldsymbol{\mu}}_t$ and $\mathbf{w}_t := \mathbf{V}_t^{1/2} \boldsymbol{\mu}_\star$. Then

$$
\max_{a \in [K]} |\widetilde{\mathbf{x}}_a^\top (\widehat{\boldsymbol{\mu}}_t - \boldsymbol{\mu}_\star)| \leq t^{-1/2} \|\widehat{\mathbf{w}}_t - \mathbf{w}_t\|_2.
$$

To use Lemma 3, we prove a bound for

$$
\left\| \sum_{\tau=1}^{t} \sum_{a \in [K]} (\widetilde{r}_{a,\tau} - \widetilde{\mathbf{x}}_a^\top \mathbf{V}_t^{-1/2} \mathbf{w}_t) \mathbf{V}_t^{-1/2} \widetilde{\mathbf{x}}_a \right\|_\infty .
$$

Let $\check{\mathbf{w}}_t^L := \mathbf{V}_t^{1/2}\check{\boldsymbol{\mu}}_t^L$. By definition of $\widetilde{r}_{a,\tau}$,

$$\left\|\sum_{\tau=1}^{t}\sum_{a\in[K]}(\widetilde{r}_{a,\tau}-\widetilde{\mathbf{x}}_a^\top\mathbf{V}_t^{-1/2}\mathbf{w}_t)\mathbf{V}_t^{-1/2}\widetilde{\mathbf{x}}_a\right\|_\infty .$$

$$= \left\|\sum_{\tau=1}^{t}\sum_{a\in[K]}\left(1-\frac{\mathbb{I}(\widetilde{a}_\tau=a)}{\phi_{a,\tau}}\right)\mathbf{V}_t^{-1/2}\widetilde{\mathbf{x}}_a\widetilde{\mathbf{x}}_a^\top\mathbf{V}_t^{-1/2}\left(\check{\mathbf{w}}_t^L-\mathbf{w}_t\right)+\frac{\mathbb{I}(\widetilde{a}_\tau=a)}{\phi_{a,\tau}}\left(y_{a,\tau}-\widetilde{\mathbf{x}}_a^\top\mathbf{V}_t^{-1/2}\mathbf{w}_t\right)\mathbf{V}_t^{-1/2}\widetilde{\mathbf{x}}_a\right\|_\infty$$

$$\le \left\|\sum_{\tau=1}^{t}\sum_{a\in[K]}\left(1-\frac{\mathbb{I}(\widetilde{a}_\tau=a)}{\phi_{a,\tau}}\right)\mathbf{V}_t^{-1/2}\widetilde{\mathbf{x}}_a\widetilde{\mathbf{x}}_a^\top\mathbf{V}_t^{-1/2}\left(\check{\mathbf{w}}_t^L-\mathbf{w}_t\right)\right\|_\infty$$

$$+ \left\|\sum_{\tau=1}^{t}\sum_{a\in[K]}\frac{\mathbb{I}(\widetilde{a}_\tau=a)}{\phi_{a,\tau}}\left(y_{a,\tau}-\widetilde{\mathbf{x}}_a^\top\mathbf{V}_t^{-1/2}\mathbf{w}_t\right)\mathbf{V}_t^{-1/2}\widetilde{\mathbf{x}}_a\right\|_\infty$$

With probability at least $1-\delta$, the event $\mathcal{M}_\tau$ happens for all $\tau\ge 1$ and we obtain a pair of matching sample $\widetilde{a}_\tau$ and $a_\tau$. Thus, the second term is equal to,

$$\left\|\sum_{\tau=1}^{t}\sum_{a\in[K]}\frac{\mathbb{I}(\widetilde{a}_\tau=a)}{\phi_{a,\tau}}\left(y_{a,\tau}-\widetilde{\mathbf{x}}_a^\top\mathbf{V}_t^{-1/2}\mathbf{w}_t\right)\mathbf{V}_t^{-1/2}\widetilde{\mathbf{x}}_a\right\|_\infty = \left\|\sum_{\tau=1}^{t}\sum_{a\in[K]}\frac{\mathbb{I}(a_\tau=a)}{\phi_{a,\tau}}\left(y_{a,\tau}-\widetilde{\mathbf{x}}_a^\top\boldsymbol{\mu}_\star\right)\mathbf{V}_t^{-1/2}\widetilde{\mathbf{x}}_a\right\|_\infty$$

$$= \frac{1}{p}\left\|\sum_{\tau=1}^{t}\epsilon_{a,\tau}\mathbf{V}_t^{-1/2}\widetilde{\mathbf{x}}_{a_\tau}\right\|_\infty .$$

Because $\|\mathbf{v}\|_\infty = \max_{i\in[d]}|e_i^\top\mathbf{v}|$ for any $\mathbf{v}\in\mathbb{R}^d$,

$$\frac{1}{p}\left\|\sum_{\tau=1}^{t}\epsilon_{a,\tau}\mathbf{V}_t^{-1/2}\widetilde{\mathbf{x}}_{a_\tau}\right\|_\infty = \frac{1}{p}\max_{a\in[K]}|\sum_{\tau=1}^{t}\epsilon_{a,\tau}\mathbf{e}_a^\top\mathbf{V}_t^{-1/2}\widetilde{\mathbf{x}}_{a_\tau}|$$

Applying Lemma 1, with probability at least $1-\delta/t^2$,

$$\max_{a\in[K]}|\sum_{\tau=1}^{t}\epsilon_{a,\tau}\mathbf{e}_a^\top\mathbf{V}_t^{-1/2}\widetilde{\mathbf{x}}_{a_\tau}| \le \max_{a\in[K]}\sigma\sqrt{2\sum_{\tau=1}^{t}\left(\mathbf{e}_a^\top\mathbf{V}_t^{-1/2}\widetilde{\mathbf{x}}_{a_\tau}\right)^2\log\frac{2Kt^2}{\delta}}$$

$$= \max_{a\in[K]}\sigma\sqrt{2\mathbf{e}_a^\top\mathbf{V}_t^{-1/2}\left(\sum_{\tau=1}^{t}\widetilde{\mathbf{x}}_{a_\tau}\widetilde{\mathbf{x}}_{a_\tau}^\top\right)\mathbf{V}_t^{-1/2}\mathbf{e}_a\log\frac{2Kt^2}{\delta}}$$

$$\le \max_{a\in[K]}\sigma\sqrt{2\mathbf{e}_a^\top\mathbf{V}_t^{-1/2}\left(\mathbf{V}_t\right)\mathbf{V}_t^{-1/2}\mathbf{e}_a\log\frac{2Kt^2}{\delta}}$$

$$= \sigma\sqrt{2\log\frac{2Kt^2}{\delta}},$$

and thus,

$$\frac{1}{\sqrt{p}}\left\|\sum_{\tau=1}^{t}\epsilon_{a,\tau}\mathbf{V}_t^{-1/2}\widetilde{\mathbf{x}}_{a_\tau}\right\|_\infty \le \sigma\sqrt{\frac{2}{p}\log\frac{2Kt^2}{\delta}} \tag{18}$$

Let $\mathbf{A}_t := \sum_{\tau=1}^{t}\sum_{a\in[K]}\frac{\mathbb{I}(\widetilde{a}_\tau=a)}{\phi_{a,\tau}}\widetilde{\mathbf{x}}_a\widetilde{\mathbf{x}}_a^\top$. Then the first term,

$$\left\|\sum_{\tau=1}^{t}\sum_{a\in[K]}\left(1-\frac{\mathbb{I}(\widetilde{a}_\tau=a)}{\phi_{a,\tau}}\right)\mathbf{V}_t^{-1/2}\widetilde{\mathbf{x}}_a\widetilde{\mathbf{x}}_a^\top\mathbf{V}_t^{-1/2}\left(\check{\mathbf{w}}_t^L-\mathbf{w}_t\right)\right\|_\infty \tag{19}$$

$$= \left\|\mathbf{V}_t^{-1/2}\left(\mathbf{V}_t-\mathbf{A}_t\right)\mathbf{V}_t^{-1/2}\left(\check{\mathbf{w}}_t^L-\mathbf{w}_t\right)\right\|_\infty .$$

Since $\|\mathbf{v}\|_\infty = \max_{i \in [d]} |\mathbf{e}_i^\top \mathbf{v}|$ for any $\mathbf{v} \in \mathbb{R}^d$,

$$\left\| \mathbf{V}_t^{-1/2} \left( \mathbf{V}_t - \mathbf{A}_t \right) \mathbf{V}_t^{-1/2} \left( \check{\mathbf{w}}_t^L - \mathbf{w}_t \right) \right\|_\infty = \max_{a \in [K]} |\mathbf{e}_a^\top \mathbf{V}_t^{-1/2} \left( \mathbf{V}_t - \mathbf{A}_t \right) \mathbf{V}_t^{-1/2} \left( \check{\mathbf{w}}_t^L - \mathbf{w}_t \right) |$$

$$\leq \max_{a \in [K]} \left\| \mathbf{e}_a^\top \mathbf{V}_t^{-1/2} \left( \mathbf{V}_t - \mathbf{A}_t \right) \mathbf{V}_t^{-1/2} \right\|_2 \|\check{\mathbf{w}}_t^L - \mathbf{w}_t\|_2 .$$

Because $\widehat{\mathbf{w}}_t$ is a minimizer of Eq. (8), by Lemma 3 and Eq. (18),

$$\left\| \check{\mathbf{w}}_t^L - \mathbf{w}_t \right\|_{\mathbf{V}_t^{-1/2} \frac{1}{p} \sum_{\tau=1}^t \widetilde{\mathbf{x}}_{a_\tau} \widetilde{\mathbf{x}}_{a_\tau}^\top \mathbf{V}_t^{-1/2}} \leq 4\sigma \sqrt{\frac{2(d + d_h) \log \frac{2Kt^2}{\delta}}{p\lambda_{\min} \left( \mathbf{V}_t^{-1/2} \frac{1}{p} \sum_{\tau=1}^t \widetilde{\mathbf{x}}_{a_\tau} \widetilde{\mathbf{x}}_{a_\tau}^\top \mathbf{V}_t^{-1/2} \right)}} .$$

Because $\phi_{a_\tau, \tau} = p$ and the coupling event $\cap_{\tau \geq 1} \mathcal{M}_\tau$ holds with probability at least $1 - \delta/t^2$,

$$\sum_{\tau=1}^t \frac{1}{p} \widetilde{\mathbf{x}}_{a_\tau} \widetilde{\mathbf{x}}_{a_\tau}^\top = \sum_{\tau=1}^t \sum_{a \in [K]} \frac{\mathbb{I}(a_\tau = a)}{p} \widetilde{\mathbf{x}}_a \widetilde{\mathbf{x}}_a^\top$$

$$= \sum_{\tau=1}^t \sum_{a \in [K]} \frac{\mathbb{I}(a_\tau = a)}{\phi_{a,\tau}} \widetilde{\mathbf{x}}_a \widetilde{\mathbf{x}}_a^\top$$

$$= \sum_{\tau=1}^t \sum_{a \in [K]} \frac{\mathbb{I}(\widetilde{a}_\tau = a)}{\phi_{a,\tau}} \widetilde{\mathbf{x}}_a \widetilde{\mathbf{x}}_a^\top$$

$$:= \mathbf{A}_t .$$

Thus, under the coupling event $\cap_{\tau=1}^t \mathcal{M}_\tau$,

$$\left\| \check{\mathbf{w}}_t^L - \mathbf{w}_t \right\|_{\mathbf{V}_t^{-1/2} \mathbf{A}_t \mathbf{V}_t^{-1/2}} \leq 4\sigma \sqrt{\frac{2(d + d_h) \log \frac{2Kt^2}{\delta}}{p\lambda_{\min} \left( \mathbf{V}_t^{-1/2} \mathbf{A}_t \mathbf{V}_t^{-1/2} \right)}} .$$

By Corollary 1, with $\epsilon \in (0, 1)$ to be determined later, for $t \geq 8\epsilon^{-2}(1-p)^{-2}K^2 \log \frac{2dt^2}{\delta}$, with probability at least $1 - \delta/t^2$,

$$\left\| \mathbf{I}_K - \mathbf{V}_t^{-1/2} \mathbf{A}_t \mathbf{V}_t^{-1/2} \right\|_2 \leq \epsilon, \tag{20}$$

which implies, $(1 - \epsilon)\mathbf{I}_K \preceq \mathbf{V}_t^{-1/2} \mathbf{A}_t \mathbf{V}_t^{-1/2}$ Thus,

$$\left\| \check{\mathbf{w}}_t^L - \mathbf{w}_t \right\|_2 \leq \frac{4\sigma}{1 - \epsilon} \sqrt{\frac{2(d + d_h) \log \frac{2Kt^2}{\delta}}{p}} .$$

Now Eq. (19) is bounded by,

$$\left\| \mathbf{V}_t^{-1/2} \left( \mathbf{V}_t - \mathbf{A}_t \right) \mathbf{V}_t^{-1/2} \left( \check{\mathbf{w}}_t^L - \mathbf{w}_t \right) \right\|_\infty$$

$$\leq \max_{a \in [K]} \left\| \mathbf{e}_a^\top \mathbf{V}_t^{-1/2} \left( \mathbf{V}_t - \mathbf{A}_t \right) \mathbf{V}_t^{-1/2} \right\|_2 \frac{4\sigma}{1 - \epsilon} \sqrt{\frac{2(d + d_h) \log \frac{2Kt^2}{\delta}}{p}} . \tag{21}$$

With simple algebra,

$$\max_{a \in [K]} \left\| \mathbf{e}_a^\top \mathbf{V}_t^{-1/2} \left( \mathbf{V}_t - \mathbf{A}_t \right) \mathbf{V}_t^{-1/2} \right\|_2$$

$$= \max_{a \in [K]} \sqrt{\lambda_{\max} \left( \mathbf{V}_t^{-1/2} \left( \mathbf{V}_t - \mathbf{A}_t \right) \mathbf{V}_t^{-1/2} (e_a e_a^\top) \mathbf{V}_t^{-1/2} \left( \mathbf{V}_t - \mathbf{A}_t \right) \mathbf{V}_t^{-1/2} \right)}$$

$$\leq \max_{a \in [K]} \sqrt{\lambda_{\max} \left( \mathbf{V}_t^{-1/2} \left( \mathbf{V}_t - \mathbf{A}_t \right) \mathbf{V}_t^{-1/2} \mathbf{I}_K \mathbf{V}_t^{-1/2} \left( \mathbf{V}_t - \mathbf{A}_t \right) \mathbf{V}_t^{-1/2} \right)}$$

$$= \left\| \mathbf{V}_t^{-1/2} \left( \mathbf{V}_t - \mathbf{A}_t \right) \mathbf{V}_t^{-1/2} \right\|_2$$

$$= \left\| \mathbf{I}_K - \mathbf{V}_t^{-1/2} \mathbf{A}_t \mathbf{V}_t^{-1/2} \right\|_2 \leq \epsilon$$

Thus,

$$\left\| \mathbf{V}_t^{-1/2} \left( \mathbf{V}_t - \mathbf{A}_t \right) \mathbf{V}_t^{-1/2} \left( \check{\mathbf{w}}_t^L - \mathbf{w}_t \right) \right\|_\infty \leq \frac{4\sigma\epsilon}{1-\epsilon} \sqrt{\frac{2(d+d_h)\log\frac{2Kt^2}{\delta}}{p}}.$$

Now we obtain,

$$\left\| \sum_{\tau=1}^t \sum_{a\in[K]} (\widetilde{r}_{a,\tau} - \widetilde{\mathbf{x}}_a^\top \mathbf{V}_t^{-1/2}\mathbf{w}_t)\mathbf{V}_t^{-1/2}\widetilde{\mathbf{x}}_a \right\|_\infty \leq \frac{4\sigma\epsilon}{1-\epsilon}\sqrt{\frac{2(d+d_h)\log\frac{2Kt^2}{\delta}}{p}} + \frac{\sigma}{p}\sqrt{2\log\frac{2Kt^2}{\delta}}$$

$$\leq \frac{4\sigma\epsilon}{1-\epsilon}\sqrt{\frac{2K\log\frac{2Kt^2}{\delta}}{p}} + \frac{\sigma}{p}\sqrt{2\log\frac{2Kt^2}{\delta}}$$

$$= \left( \frac{4\epsilon\sqrt{K}}{1-\epsilon} + \frac{1}{p} \right) \sigma\sqrt{2\log\frac{2Kt^2}{\delta}}$$

Setting $\epsilon = p^{-1}/(\sqrt{K}+p^{-1})$ gives $\frac{\epsilon\sqrt{K}}{1-\epsilon} = p^{-1}$ and for $t \geq 8(\sqrt{K}+p^{-1})^2p^2(1-p)^{-2}K^2\log\frac{2Kt^2}{\delta}$,

$$\left\| \sum_{\tau=1}^t \sum_{a\in[K]} (\widetilde{r}_{a,\tau} - \widetilde{\mathbf{x}}_a^\top \mathbf{V}_t^{-1/2}\mathbf{w}_t)\mathbf{V}_t^{-1/2}\widetilde{\mathbf{x}}_a \right\|_\infty \leq \frac{5\sigma}{p}\sqrt{2\log\frac{2Kt^2}{\delta}}$$

Because $\widehat{\mathbf{w}}_t$ is a minimizer of (11), by Lemma 3,

$$\|\widehat{\mathbf{w}}_t - \mathbf{w}_t\|_{\mathbf{V}_t^{-1/2}(\sum_{\tau=1}^t \widetilde{\mathbf{x}}_a\widetilde{\mathbf{x}}_a)\mathbf{V}_t^{-1/2}} \leq \frac{20\sigma}{p}\sqrt{\frac{2(d+d_h)\log\frac{2Kt^2}{\delta}}{\lambda_{\min}\left(\mathbf{V}_t^{-1/2}(\sum_{\tau=1}^t \widetilde{\mathbf{x}}_a\widetilde{\mathbf{x}}_a)\mathbf{V}_t^{-1/2}\right)}},$$

which is equivalent to,

$$\|\widehat{\mathbf{w}}_t - \mathbf{w}_t\|_2 \leq \frac{20\sigma}{p}\sqrt{2(d+d_h)\log\frac{2Kt^2}{\delta}}.$$

This concludes,

$$\max_{a\in[K]} |\widetilde{\mathbf{x}}_a(\widehat{\boldsymbol{\mu}}_t - \boldsymbol{\mu}_\star)| \leq \frac{20\sigma}{p}\sqrt{\frac{2(d+d_h)\log\frac{2Kt^2}{\delta}}{t}},$$

which conmpletes the proof. $\qquad\qquad\square$

### C.3 PROOF OF THEOREM 3

Because the regret is bounded by 2 and the number of rounds for the exploration phase is at most $|\mathcal{E}_T| \leq 8(\sqrt{K}+p^{-1})^2p^2(1-p)^{-2}K^2\log\frac{2KT^2}{\delta}$.

$$\text{Reg}(T) \leq \frac{16K^2(\sqrt{K}+p^{-1})^2}{(1-p)^2}\log\frac{2Kt^2}{\delta} + \sum_{t\in[T]\backslash\mathcal{E}_T} \mathbb{E}_{t-1}[y_{\star,t}] - \mathbb{E}_{t-1}[y_{a_t,t}]$$

$$= \frac{16K^2(\sqrt{K}+p^{-1})^2}{(1-p)^2} + \sum_{t\in[T]\backslash\mathcal{E}_T} \{\mathbb{I}(a_t = \widehat{a}_t)(\mathbb{E}_{t-1}[y_{\star,t}] - \mathbb{E}_{t-1}[y_{a_t,t}])\}$$

$$+ \sum_{t\in[T]\backslash\mathcal{E}_T} \{\mathbb{I}(a_t \neq \widehat{a}_t)(\mathbb{E}_{t-1}[y_{\star,t}] - \mathbb{E}_{t-1}[y_{a_t,t}])\}.$$

By Theorem 2, on the event $\{a_t = \widehat{a}_t\}$,

$$\mathbb{E}_{t-1}[y_{\star,t}] - \mathbb{E}_{t-1}[y_{a_t,t}] = \widetilde{\boldsymbol{x}}_{a_\star}^\top \boldsymbol{\mu}_\star - \widetilde{\boldsymbol{x}}_{\widehat{a}_t}^\top \boldsymbol{\mu}_\star$$

$$\leq 2\max_{a\in[K]} \left| \widetilde{\boldsymbol{x}}_a^\top \left( \boldsymbol{\mu}_\star - \widehat{\boldsymbol{\mu}}_{t-1}^L \right) \right| + \widetilde{\boldsymbol{x}}_{a_\star}^\top \widehat{\boldsymbol{\mu}}_{t-1}^L - \widetilde{\boldsymbol{x}}_{\widehat{a}_t}^\top \widehat{\boldsymbol{\mu}}_{t-1}^L$$

$$\leq 2\max_{a\in[K]} \left| \widetilde{\boldsymbol{x}}_a^\top \left( \boldsymbol{\mu}_\star - \widehat{\boldsymbol{\mu}}_{t-1}^L \right) \right|$$

$$\leq \frac{40\sigma}{p}\sqrt{\frac{2(d+d_h)}{t}\log\frac{2Kt^2}{\delta}},$$

with probability at least $1 - 5\delta/t^2$ for each $t \in [T] \setminus \mathcal{E}_T$. Summing over $t$ gives,

$$\sum_{t \in [T] \setminus \mathcal{E}_T} \{\mathbb{I}(a_t = \hat{a}_t)(\mathbb{E}_{t-1}[y_{\star,t}] - \mathbb{E}_{t-1}[y_{a_t,t}])\} \le \frac{80\sigma}{p}\sqrt{\frac{2(d+d_h)}{t}\log\frac{2Kt^2}{\delta}}.$$

By resampling at most $\rho_t$ times, the probability of the event $\{a_t \ne \hat{a}_t\}$ is

$$\mathbb{P}(a_t \ne \hat{a}_t) = \sum_{m=1}^{\rho_t} \frac{p}{(K-1)\sqrt{t}}\left(1 - \frac{p}{(K-1)\sqrt{t}}\right)^{m-1}$$

$$= \frac{p}{(K-1)\sqrt{t}}\left(\frac{p}{(K-1)\sqrt{t}}\right)^{-1}\left\{1 - \left(1 - \frac{p}{(K-1)\sqrt{t}}\right)^{\rho_t}\right\}$$

$$= 1 - \left(1 - \frac{p}{(K-1)\sqrt{t}}\right)^{\rho_t}$$

$$\ge \frac{p\rho_t}{(K-1)\sqrt{t}},$$

where the last inequality uses $(1+x)^n \ge 1 + nx$ for $x \ge -1$ and $n \in \mathbb{N}$. Then the expected sum of regret,

$$\mathbb{E}\left[\sum_{t \in [T] \setminus \mathcal{E}_T} \{\mathbb{I}(a_t = \hat{a}_t)(\mathbb{E}_{t-1}[y_{\star,t}] - \mathbb{E}_{t-1}[y_{a_t,t}])\}\right]$$

$$\le \sum_{t \in [T] \setminus \mathcal{E}_T} \mathbb{P}(a_t \ne \hat{a}_t)$$

$$\le \frac{2p\sqrt{T}}{K-1}\rho_T$$

$$= \frac{2p\sqrt{T}}{K-1}\frac{\log\frac{(T+1)^2}{\delta}}{\log(1/p)}.$$

Thus,

$$\mathbb{E}[\text{Reg}(T)] \le 10\delta \log T + \frac{16K^2(\sqrt{K} + p^{-1})^2}{(1-p)^2} + \frac{2p\sqrt{T}}{K-1}\frac{\log\frac{(T+1)^2}{\delta}}{\log(1/p)}$$

$$+ \frac{80\sigma}{p}\sqrt{2(d+d_h)T\log\frac{2KT^2}{\delta}},$$

which concludes the proof. □

### C.4 PROOF OF THEOREM 4

Let $\widetilde{\mathbf{V}}_t := \sum_{\tau=1}^t \mathbb{I}(\mathcal{M}_\tau)\sum_{a\in[K]}\widetilde{\mathbf{x}}_a\widetilde{\mathbf{x}}_a^\top + \mathbf{I}_K$ and $\mathbf{V}_t := \sum_{\tau=1}^t \sum_{a\in[K]}\widetilde{\mathbf{x}}_a\widetilde{\mathbf{x}}_a^\top + \mathbf{I}_K$. By definition of $\widehat{\boldsymbol{\mu}}_t^R$,

$$\widetilde{\mathbf{x}}_a^\top(\widehat{\boldsymbol{\mu}}_t - \boldsymbol{\mu}_\star) = \widetilde{\mathbf{x}}_a^\top\widetilde{\mathbf{V}}_t^{-1}\left\{\sum_{\tau=1}^t \mathbb{I}(\mathcal{M}_\tau)\sum_{a\in[K]}\widetilde{\mathbf{x}}_a(\widetilde{r}_{a,\tau} - \widetilde{\mathbf{x}}_a^\top\boldsymbol{\mu}_\star) - \boldsymbol{\mu}_\star\right\}.$$

By definition of the pseudo-rewards,

$$\widetilde{r}_{a,\tau} - \widetilde{\mathbf{x}}_a^\top\boldsymbol{\mu}_\star = \left(1 - \frac{\mathbb{I}(\widetilde{a}_\tau = a)}{\phi_{a,t}}\right)\widetilde{\mathbf{x}}_a^\top(\check{\boldsymbol{\mu}}_t^R - \boldsymbol{\mu}_\star) + \frac{\mathbb{I}(\widetilde{a}_\tau = a)}{\phi_{a,\tau}}\epsilon_{a,\tau}.$$

Let $\widetilde{\mathbf{A}}_t := \sum_{\tau=1}^t \mathbb{I}(\mathcal{M}_\tau)\sum_{a\in[K]}\frac{\mathbb{I}(\widetilde{a}_\tau=a)}{\phi_{a,t}}\widetilde{\mathbf{x}}_a\widetilde{\mathbf{x}}_a^\top + \mathbf{I}_K$ and $\mathbf{A}_t := \sum_{\tau=1}^t \sum_{a\in[K]}\frac{\mathbb{I}(\widetilde{a}_\tau=a)}{\phi_{a,t}}\widetilde{\mathbf{x}}_a\widetilde{\mathbf{x}}_a^\top + \mathbf{I}_K$ Then,

$$\widetilde{\mathbf{x}}_a^\top(\widehat{\boldsymbol{\mu}}_t - \boldsymbol{\mu}_\star) = \widetilde{\mathbf{x}}_a^\top\widetilde{\mathbf{V}}_t^{-1}\left\{\left(\widetilde{\mathbf{V}}_t - \widetilde{\mathbf{A}}_t\right)(\check{\boldsymbol{\mu}}_t^R - \boldsymbol{\mu}_\star) + \sum_{\tau=1}^t \mathbb{I}(\mathcal{M}_\tau)\sum_{a\in[K]}\frac{\mathbb{I}(\widetilde{a}_\tau=a)}{\phi_{a,\tau}}\widetilde{\mathbf{x}}_a\epsilon_{a,\tau} - \boldsymbol{\mu}_\star\right\}.$$

By definition of the imputation estimator $\check{\boldsymbol{\mu}}_t$,

$$
\check{\boldsymbol{\mu}}_t^R - \boldsymbol{\mu}_\star = \left( \sum_{\tau=1}^t \widetilde{\boldsymbol{x}}_{a_\tau} \widetilde{\boldsymbol{x}}_{a_\tau}^\top + p\mathbf{I}_K \right)^{-1} \left( \sum_{\tau=1}^t \widetilde{\boldsymbol{x}}_{a_\tau} \epsilon_{a_\tau,\tau} - p\boldsymbol{\mu}_\star \right)
$$

$$
= \left( \sum_{\tau=1}^t \frac{1}{\phi_{a_\tau,\tau}} \widetilde{\boldsymbol{x}}_{a_\tau} \widetilde{\boldsymbol{x}}_{a_\tau}^\top + \mathbf{I}_K \right)^{-1} \left( \sum_{\tau=1}^t \frac{1}{\phi_{a_\tau,\tau}} \widetilde{\boldsymbol{x}}_{a_\tau} \epsilon_{a_\tau,\tau} - \boldsymbol{\mu}_\star \right)
$$

$$
= \left( \sum_{\tau=1}^t \sum_{a \in [K]} \frac{\mathbb{I}(\widetilde{a}_\tau = a)}{\phi_{a,\tau}} \widetilde{\boldsymbol{x}}_{a_\tau} \widetilde{\boldsymbol{x}}_{a_\tau}^\top + \mathbf{I}_K \right)^{-1} \left( \sum_{\tau=1}^t \frac{1}{p} \widetilde{\boldsymbol{x}}_{a_\tau} \epsilon_{a_\tau,\tau} - \boldsymbol{\mu}_\star \right),
$$

where the second equality holds because $\phi_{a_\tau,\tau} = p$. Under the coupling event $\cap_{\tau=1}^t \mathcal{M}_\tau$,

$$
\sum_{\tau=1}^t \mathbb{I}(\mathcal{M}_\tau) \sum_{a \in [K]} \frac{\mathbb{I}(\widetilde{a}_\tau = a)}{\phi_{a,t}} \widetilde{\mathbf{x}}_a \widetilde{\mathbf{x}}_a^\top + \mathbf{I}_K = \sum_{\tau=1}^t \sum_{a \in [K]} \frac{\mathbb{I}(\widetilde{a}_\tau = a)}{\phi_{a,t}} \widetilde{\mathbf{x}}_a \widetilde{\mathbf{x}}_a^\top + \mathbf{I}_K
$$

$$
:= \mathbf{A}_t,
$$

and

$$
\widetilde{\mathbf{x}}_a^\top (\widehat{\boldsymbol{\mu}}_t - \boldsymbol{\mu}_\star) = \widetilde{\mathbf{x}}_a^\top \mathbf{V}_t^{-1} \left\{ (\mathbf{V}_t - \mathbf{A}_t) \mathbf{A}_t^{-1} \left( \sum_{\tau=1}^t \frac{1}{p} \widetilde{\boldsymbol{x}}_{a_\tau} \epsilon_{a_\tau,\tau} - \boldsymbol{\mu}_\star \right) + \sum_{\tau=1}^t \frac{\epsilon_{a_\tau}}{\phi_{a_\tau,\tau}} \widetilde{\boldsymbol{x}}_{a_\tau} - \boldsymbol{\mu}_\star \right\}
$$

$$
= \widetilde{\mathbf{x}}_a^\top \mathbf{V}_t^{-1} \left\{ (\mathbf{V}_t - \mathbf{A}_t) \mathbf{A}_t^{-1} + \mathbf{I}_K \right\} \left( \sum_{\tau=1}^t \frac{1}{p} \widetilde{\boldsymbol{x}}_{a_\tau} \epsilon_{a_\tau,\tau} - \boldsymbol{\mu}_\star \right)
$$

$$
= \widetilde{\mathbf{x}}_a^\top \mathbf{V}_t^{-1/2} \left( \mathbf{V}_t^{1/2} \mathbf{A}_t^{-1} \mathbf{V}_t^{1/2} \right) \mathbf{V}_t^{-1/2} \left( \sum_{\tau=1}^t \frac{1}{p} \widetilde{\boldsymbol{x}}_{a_\tau} \epsilon_{a_\tau,\tau} - \boldsymbol{\mu}_\star \right).
$$

Taking absolute value on both sides,

$$
\max_{a \in [K]} |\widetilde{\mathbf{x}}_a^\top (\widehat{\boldsymbol{\mu}}_t - \boldsymbol{\mu}_\star)| \leq \max_{a \in [K]} \|\widetilde{\mathbf{x}}_a\|_{\mathbf{V}_t^{-1}} \|\mathbf{V}_t^{1/2} \mathbf{A}_t^{-1} \mathbf{V}_t^{1/2}\|_2 \left\| \sum_{\tau=1}^t \frac{1}{p} \widetilde{\boldsymbol{x}}_{a_\tau} \epsilon_{a_\tau,\tau} - \boldsymbol{\mu}_\star \right\|_{\mathbf{V}_t^{-1}}.
$$

By Corollary which implies $\mathbf{I}_K - \mathbf{V}_t^{-1/2} \mathbf{A}_t \mathbf{V}_t^{-1/2} \preceq \epsilon \mathbf{I}_K$. Rearraging the terms,

$$
\mathbf{V}_t^{1/2} \mathbf{A}_t^{-1} \mathbf{V}_t^{1/2} \preceq (1 - \epsilon)^{-1} \mathbf{I}_K.
$$

Thus,

$$
\max_{a \in [K]} |\widetilde{\mathbf{x}}_a^\top (\widehat{\boldsymbol{\mu}}_t - \boldsymbol{\mu}_\star)| \leq \frac{\max_{a \in [K]} \|\widetilde{\mathbf{x}}_a\|_{\mathbf{V}_t^{-1}}}{1 - \epsilon} \left\| \sum_{\tau=1}^t \frac{1}{p} \widetilde{\boldsymbol{x}}_{a_\tau} \epsilon_{a_\tau,\tau} - \boldsymbol{\mu}_\star \right\|_{\mathbf{V}_t^{-1}}
$$

$$
\leq \frac{\max_{a \in [K]} \|\widetilde{\mathbf{x}}_a\|_{\mathbf{V}_t^{-1}}}{1 - \epsilon} \left( \frac{1}{p} \left\| \sum_{\tau=1}^t \widetilde{\boldsymbol{x}}_{a_\tau} \epsilon_{a_\tau,\tau} \right\|_{\mathbf{V}_t^{-1}} + \|\boldsymbol{\mu}_\star\|_{\mathbf{V}_t^{-1}} \right).
$$

Note that the matrix $\mathbf{V}_t$ is deterministic. By Lemma 9 in (Abbasi-yadkori et al., 2011), with probability at least $1 - \delta$,

$$
\left\| \sum_{\tau=1}^t \widetilde{\boldsymbol{x}}_{a_\tau} \epsilon_{a_\tau,\tau} \right\|_{\mathbf{V}_t^{-1}} \leq \left\| \sum_{\tau=1}^t \widetilde{\boldsymbol{x}}_{a_\tau} \epsilon_{a_\tau,\tau} \right\|_{\left( \sum_{\tau=1}^t \widetilde{\boldsymbol{x}}_{a_\tau} \widetilde{\boldsymbol{x}}_{a_\tau}^\top + \mathbf{I}_K \right)^{-1}}
$$

$$
\leq \sigma \sqrt{2 \log \frac{\det(\sum_{\tau=1}^t \widetilde{\boldsymbol{x}}_{a_\tau} \widetilde{\boldsymbol{x}}_{a_\tau}^\top + \mathbf{I}_K)^{1/2}}{\delta}}
$$

$$
\leq \sigma \sqrt{\log \frac{\det(\sum_{\tau=1}^t \widetilde{\boldsymbol{x}}_{a_\tau} \widetilde{\boldsymbol{x}}_{a_\tau}^\top + \mathbf{I}_K)}{\delta}},
$$

for all $t \geq 1$. Because

$$\det \left( \sum_{\tau=1}^{t} \widetilde{\boldsymbol{x}}_{a_\tau} \widetilde{\boldsymbol{x}}_{a_\tau}^\top + \mathbf{I}_K \right) \leq \left\{ \frac{\operatorname{Tr}\left( \sum_{\tau=1}^{t} \widetilde{\boldsymbol{x}}_{a_\tau} \widetilde{\boldsymbol{x}}_{a_\tau}^\top \right) + K}{K} \right\}^K$$

$$\leq \left\{ \frac{t \max_{a \in [K]} \|\widetilde{\boldsymbol{x}}_{a_\tau}\|_2 + K}{K} \right\}^K$$

$$\leq \{t+1\}^K,$$

where the last inequality holds by $\|\widetilde{\boldsymbol{x}}_{a_\tau}\|_2 \leq \sqrt{K}\|\widetilde{\boldsymbol{x}}_{a_\tau}\|_\infty \leq K$. Thus,

$$\left\| \sum_{\tau=1}^{t} \widetilde{\boldsymbol{x}}_{a_\tau} \epsilon_{a_\tau,\tau} \right\|_{\mathbf{V}_t^{-1}} \leq \sigma \sqrt{K \log \frac{t+1}{\delta}},$$

which proves,

$$\max_{a \in [K]} |\widetilde{\mathbf{x}}_a^\top (\widehat{\boldsymbol{\mu}}_t - \boldsymbol{\mu}_\star)| \leq \frac{\max_{a \in [K]} \|\widetilde{\mathbf{x}}_a\|_{\mathbf{V}_t^{-1}}}{1-\epsilon} \left( \frac{\sigma}{p} \sqrt{K \log \frac{t+1}{\delta}} + \|\boldsymbol{\mu}_\star\|_{\mathbf{V}_t^{-1}} \right)$$

$$\leq \frac{1}{\sqrt{t}} \cdot \frac{1}{1-\epsilon} \left( \frac{\sigma}{p} \sqrt{K \log \frac{t+1}{\delta}} + \|\boldsymbol{\mu}_\star\|_{\mathbf{V}_t^{-1}} \right).$$

Because $\|\boldsymbol{\mu}_\star\|_{\mathbf{V}_t^{-1}} \leq \|\boldsymbol{\mu}_\star\|_2 \leq \sqrt{K}$, setting $\epsilon = 1/2$ completes the proof. □

## C.5 PROOF OF THEOREM 5

Because the regret is bounded by 1 and the number of rounds for the exploration phase is at most $|\mathcal{E}_T| \leq 32(1-p)^{-2} K^2 \log \frac{2dT^2}{\delta}$.

$$\operatorname{Reg}(T) \leq 32(1-p)^{-2} K^2 \log \frac{2dT^2}{\delta} + \sum_{t \in [T] \setminus \mathcal{E}_T} \mathbb{E}_{t-1}[y_{\star,t}] - \mathbb{E}_{t-1}[y_{a_t,t}]$$

$$= 32(1-p)^{-2} K^2 \log \frac{2dT^2}{\delta} + \sum_{t \in [T] \setminus \mathcal{E}_T} \left\{ \mathbb{I}\left(a_t = \widehat{a}_t\right) \left( \mathbb{E}_{t-1}[y_{\star,t}] - \mathbb{E}_{t-1}[y_{a_t,t}] \right) \right\}$$

$$+ \sum_{t \in [T] \setminus \mathcal{E}_T} \left\{ \mathbb{I}\left(a_t \neq \widehat{a}_t\right) \left( \mathbb{E}_{t-1}[y_{\star,t}] - \mathbb{E}_{t-1}[y_{a_t,t}] \right) \right\}.$$

On the event $\{a_t = \widehat{a}_t\}$,

$$\mathbb{E}_{t-1}[y_{\star,t}] - \mathbb{E}_{t-1}[y_{a_t,t}] = \widetilde{\boldsymbol{x}}_{a_\star}^\top \boldsymbol{\mu}_\star - \widetilde{\boldsymbol{x}}_{\widehat{a}_t}^\top \boldsymbol{\mu}_\star$$

$$\leq 2 \max_{a \in [K]} \left| \widetilde{\mathbf{x}}_a^\top \left( \boldsymbol{\mu}_\star - \widehat{\boldsymbol{\mu}}_{t-1}^R \right) \right| + \widetilde{\boldsymbol{x}}_{a_\star}^\top \widehat{\boldsymbol{\mu}}_{t-1}^R - \widetilde{\boldsymbol{x}}_{\widehat{a}_t}^\top \widehat{\boldsymbol{\mu}}_{t-1}^R$$

$$\leq 2 \max_{a \in [K]} \left| \widetilde{\mathbf{x}}_a^\top \left( \boldsymbol{\mu}_\star - \widehat{\boldsymbol{\mu}}_{t-1}^R \right) \right|$$

$$\leq \frac{4}{\sqrt{t}} \left( \frac{\sigma}{p} \sqrt{K \log \frac{2t^2}{\delta}} + \sqrt{K} \right),$$

with probability at least $1 - 3\delta/t$, by Theorem 4. Summing over $t$ gives,

$$\sum_{t \in [T] \setminus \mathcal{E}_T} \left\{ \mathbb{I}\left(a_t = \widehat{a}_t\right) \left( \mathbb{E}_{t-1}[y_{\star,t}] - \mathbb{E}_{t-1}[y_{a_t,t}] \right) \right\} \leq 8\sqrt{KT} \left( \frac{\sigma}{p} \sqrt{\log \frac{2T^2}{\delta}} + 1 \right).$$

By resampling at most $\rho_t$ times, the probability of the event $\{a_t \neq \widehat{a}_t\}$ is

$$
\begin{aligned}
\mathbb{P}\left(a_t \neq \widehat{a}_t\right) &= \sum_{m=1}^{\rho_t} \frac{p}{(K-1)\sqrt{t}} \left(1 - \frac{p}{(K-1)\sqrt{t}}\right)^{m-1} \\
&= \frac{p}{(K-1)\sqrt{t}} \left(\frac{p}{(K-1)\sqrt{t}}\right)^{-1} \left\{1 - \left(1 - \frac{p}{(K-1)\sqrt{t}}\right)^{\rho_t}\right\} \\
&= 1 - \left(1 - \frac{p}{(K-1)\sqrt{t}}\right)^{\rho_t} \\
&\geq \frac{p\rho_t}{(K-1)\sqrt{t}},
\end{aligned}
$$

where the last inequality uses $(1+x)^n \geq 1 + nx$ for $x \geq -1$ and $n \in \mathbb{N}$. Then the expected sum of regret,

$$
\begin{aligned}
\mathbb{E}\left[\sum_{t \in [T] \setminus \mathcal{E}_T} \left\{\mathbb{I}\left(a_t = \widehat{a}_t\right)\left(\mathbb{E}_{t-1}[y_{\star,t}] - \mathbb{E}_{t-1}[y_{a_t,t}]\right)\right\}\right] &\leq \sum_{t \in [T] \setminus \mathcal{E}_T} \mathbb{P}\left(a_t \neq \widehat{a}_t\right) \\
&\leq \frac{2p\sqrt{T}}{K-1}\rho_T \\
&= \frac{2p\sqrt{T}}{K-1} \frac{\log \frac{(T+1)^2}{\delta}}{\log(1/p)}.
\end{aligned}
$$

Thus,

$$
\mathbb{E}[\mathrm{Reg}(T)] \leq 6\delta \log T + \frac{32K^2}{(1-p)^2} \log \frac{2dT^2}{\delta} + \frac{2p\sqrt{T}}{K-1} \frac{\log \frac{(T+1)^2}{\delta}}{\log(1/p)} + 8\sqrt{KT}\left(\frac{\sigma}{p}\sqrt{\log \frac{T+1}{\delta}} + 1\right). \quad \square
$$

## D    TECHNICAL LEMMAS

**Lemma 1.** *(Exponential martingale inequality) If a martingale $(\mathbf{X}_t; t \geq 0)$, adapted to filtration $\mathcal{F}_t$, satisfies $\mathbb{E}[\exp(\lambda \mathbf{X}_t)|\mathcal{F}_{t-1}] \leq \exp(\lambda^2 \sigma_t^2/2)$ for some constant $\sigma_t$, for all $t$, then for any $a \geq 0$,*

$$
\mathbb{P}\left(|X_T - X_0| \geq a\right) \leq 2\exp\left(-\frac{a^2}{2\sum_{t=1}^{T}\sigma_t^2}\right)
$$

*Thus, with probability at least $1 - \delta$,*

$$
|X_T - X_0| \leq \sqrt{2\sum_{t=1}^{T}\sigma_t^2 \log \frac{2}{\delta}}.
$$

### D.1    A HOEFFDING BOUND FOR MATRICES

**Lemma 2.** *Let $\{\mathbf{M}_\tau : \tau \in [t]\}$ be a $\mathbb{R}^{d \times d}$-valued stochastic process adapted to the filtration $\{\mathcal{F}_\tau : \tau \in [t]\}$, i.e., $\mathbf{M}_\tau$ is $\mathcal{F}_\tau$-measurable for $\tau \in [t]$. Suppose that the matrix $\mathbf{M}_\tau$ is symmetric and the eigenvalues of the difference $\mathbf{M}_\tau - \mathbb{E}[\mathbf{M}_\tau | \mathcal{F}_{\tau-1}]$ lie in $[-b, b]$ for some $b > 0$. Then for $x > 0$,*

$$
\mathbb{P}\left(\left\|\sum_{\tau=1}^{t} \mathbf{M}_\tau - \mathbb{E}[\mathbf{M}_\tau | \mathcal{F}_{\tau-1}]\right\|_2 \geq x\right) \leq 2d\exp\left(-\frac{x^2}{2tb^2}\right)
$$

*Proof.* The proof is an adapted version of Hoeffding's inequality for matrix stochastic process with the argument of (Tropp, 2012). Let $\mathbf{D}_\tau := \mathbf{M}_\tau - \mathbb{E}[\mathbf{M}_\tau | \mathcal{F}_{\tau-1}]$. Then, for $x > 0$,

$$
\mathbb{P}\left(\left\|\sum_{\tau=1}^{t} \mathbf{D}_\tau\right\|_2 \geq x\right) \leq \mathbb{P}\left(\lambda_{\max}\left(\sum_{\tau=1}^{t} \mathbf{D}_\tau\right) \geq x\right) + \mathbb{P}\left(\lambda_{\max}\left(-\sum_{\tau=1}^{t} \mathbf{D}_\tau\right) \geq x\right)
$$

We bound the first term and the second term is bounded with similar arguement. For any $v > 0$,

$$\mathbb{P}\left(\lambda_{\max}\left(\sum_{\tau=1}^{t}\mathbf{D}_\tau\right) \geq x\right) \leq \mathbb{P}\left(\exp\left\{v\lambda_{\max}\left(\sum_{\tau=1}^{t}\mathbf{D}_\tau\right)\right\} \geq e^{vx}\right) \leq e^{-vx}\mathbb{E}\left[\exp\left\{v\lambda_{\max}\left(\sum_{\tau=1}^{t}\mathbf{D}_\tau\right)\right\}\right].$$

Since $\sum_{\tau=1}^{t}\mathbf{D}_\tau$ is a real symmetric matrix,

$$\exp\left\{v\lambda_{\max}\left(\sum_{\tau=1}^{t}\mathbf{D}_\tau\right)\right\} = \lambda_{\max}\left\{\exp\left(v\sum_{\tau=1}^{t}\mathbf{D}_\tau\right)\right\} \leq \mathrm{Tr}\left\{\exp\left(v\sum_{\tau=1}^{t}\mathbf{D}_\tau\right)\right\},$$

where the last inequality holds since $\exp(v\sum_{\tau=1}^{t}\mathbf{D}_\tau)$ has nonnegative eigenvalues. Taking expectation on both side gives,

$$\mathbb{E}\left[\exp\left\{v\lambda_{\max}\left(\sum_{\tau=1}^{t}\mathbf{D}_\tau\right)\right\}\right] \leq \mathbb{E}\left[\mathrm{Tr}\left\{\exp\left(v\sum_{\tau=1}^{t}\mathbf{D}_\tau\right)\right\}\right]$$

$$= \mathrm{Tr}\mathbb{E}\left[\exp\left(v\sum_{\tau=1}^{t}\mathbf{D}_\tau\right)\right]$$

$$= \mathrm{Tr}\mathbb{E}\left[\exp\left(v\sum_{\tau=1}^{t-1}\mathbf{D}_\tau + \log\exp(v\mathbf{D}_t)\right)\right].$$

By Lieb's theorem (Tropp, 2015) the mapping $\mathbf{D} \mapsto \exp(\mathbf{H} + \log\mathbf{D})$ is concave on positive symmetric matrices for any symmetric positive definite $H$. By Jensen's inequality,

$$\mathrm{Tr}\mathbb{E}\left[\exp\left(v\sum_{\tau=1}^{t-1}\mathbf{D}_\tau + \log\exp(v\mathbf{D}_t)\right)\right] \leq \mathrm{Tr}\mathbb{E}\left[\exp\left(v\sum_{\tau=1}^{t-1}\mathbf{D}_\tau + \log\mathbb{E}\left[\exp(v\mathbf{D}_t)|\mathcal{F}_{t-1}\right]\right)\right]$$

By Hoeffding's lemma,

$$e^{vx} \leq \frac{b-x}{2b}e^{-vb} + \frac{x+b}{2b}e^{vb}$$

for all $x \in [-b, b]$. Because the eigenvalue of $\mathbf{D}_\tau$ lies in $[-b, b]$, we have

$$\mathbb{E}\left[\exp(v\mathbf{D}_t)|\mathcal{F}_{t-1}\right] \preceq \mathbb{E}\left[\frac{e^{-vb}}{2b}(b\mathbf{I}_d - \mathbf{D}_t) + \frac{e^{vb}}{2b}(\mathbf{D}_t + b\mathbf{I}_d)\Big|\mathcal{F}_{t-1}\right]$$

$$= \frac{e^{-vb} + e^{vb}}{2}\mathbf{I}_d$$

$$\preceq \exp(\frac{v^2 b^2}{2})\mathbf{I}_d.$$

Recursively,

$$
\mathbb{E}\left[\exp\left\{v\lambda_{\max}\left(\sum_{\tau=1}^{t}\mathbf{D}_{\tau}\right)\right\}\right] \leq \mathrm{Tr}\,\mathbb{E}\left[\exp\left(v\sum_{\tau=1}^{t-1}\mathbf{D}_{\tau} + \log\mathbb{E}\left[\exp(v\mathbf{D}_{t})\,|\,\mathcal{F}_{t-1}\right]\right)\right]
$$

$$
\leq \mathrm{Tr}\,\mathbb{E}\left[\exp\left(v\sum_{\tau=1}^{t-1}\mathbf{D}_{\tau} + (\frac{v^2 b^2}{2})\mathbf{I}_d\right)\right]
$$

$$
\leq \mathrm{Tr}\,\mathbb{E}\left[\exp\left(v\sum_{\tau=1}^{t-2}\mathbf{D}_{\tau} + (\frac{v^2 b^2}{2})\mathbf{I}_d + \log\mathbb{E}\left[\exp(vD_{t-1})\,|\,\mathcal{F}_{t-2}\right]\right)\right]
$$

$$
\leq \mathrm{Tr}\,\mathbb{E}\left[\exp\left(v\sum_{\tau=1}^{t-2}\mathbf{D}_{\tau} + (\frac{2v^2 b^2}{2})\mathbf{I}_d\right)\right]
$$

$$
\vdots
$$

$$
\leq \mathrm{Tr}\exp\left((\frac{tv^2 b^2}{2})\mathbf{I}_d\right)
$$

$$
= \exp\left(\frac{tv^2 b^2}{2}\right)\mathrm{Tr}\left(\mathbf{I}_d\right)
$$

$$
= d\exp\left(\frac{tv^2 b^2}{2}\right).
$$

Thus we have

$$
\mathbb{P}\left(\lambda_{\max}\left(\sum_{\tau=1}^{t}\mathbf{D}_{\tau}\right) \geq x\right) \leq d\exp\left(-vx + \frac{tv^2 b^2}{2}\right).
$$

Minimizing over $v > 0$ gives $v = x/(tb^2)$ and

$$
\mathbb{P}\left(\lambda_{\max}\left(\sum_{\tau=1}^{t}\mathbf{D}_{\tau}\right) \geq x\right) \leq d\exp\left(-\frac{x^2}{2tb^2}\right),
$$

which proves the lemma. $\qquad\square$

### D.2 A Bound for the Gram Matrix

The Hoeffding bound for matrices (Lemma 2) implies the following bounf for the two Gram matrices $\mathbf{A}_t := \sum_{\tau=1}^{t}\widetilde{\mathbf{x}}_{a_\tau}\widetilde{\mathbf{x}}_{a_\tau}^{\top}$ and $\mathbf{V}_t := \sum_{\tau=1}^{t}\sum_{a\in[K]}\widetilde{\mathbf{x}}_a\widetilde{\mathbf{x}}_a^{\top}$

**Corollary 1.** *For any $\epsilon \in (0,1)$ and $t \geq 8\epsilon^{-2}(1-p)^{-2}K^2\log\frac{2Kt^2}{\delta}$, with probability at least $1 - \delta/t^2$,*

$$
\left\|\mathbf{I}_K - \mathbf{V}_t^{-1/2}\mathbf{A}_t\mathbf{V}_t^{-1/2}\right\|_2 \leq \epsilon,
$$

*Proof.* Note that

$$
\mathbf{V}_t^{-1/2}\mathbf{A}_t\mathbf{V}_t^{-1/2} - \mathbf{I}_K = \mathbf{V}_t^{-1/2}\left\{\sum_{\tau=1}^{t}\sum_{a\in[K]}\left(\frac{\mathbb{I}(\widetilde{a}_\tau = a)}{\phi_{a,\tau}} - 1\right)\widetilde{\mathbf{x}}_a\widetilde{\mathbf{x}}_a^{\top}\right\}\mathbf{V}_t^{-1/2},
$$

and the martingale difference matrix for each $\tau \in [t]$,

$$
\left\|\sum_{a\in[K]}\left(\frac{\mathbb{I}(\widetilde{a}_\tau = a)}{\phi_{a,\tau}} - 1\right)\mathbf{V}_t^{-1/2}\widetilde{\mathbf{x}}_a\widetilde{\mathbf{x}}_a^{\top}\mathbf{V}_t^{-1/2}\right\|_2 \leq \left(\frac{K-1}{1-p} + K - 2\right)\max_{a\in[K]}\left\|\mathbf{V}_t^{-1/2}\widetilde{\mathbf{x}}_a\widetilde{\mathbf{x}}_a^{\top}\mathbf{V}_t^{-1/2}\right\|_2
$$

$$
\leq \frac{2K}{1-p}\max_{a\in[K]}\|\widetilde{\mathbf{x}}_a\|_{\mathbf{V}_t^{-1}}^2
$$

$$
\leq \frac{2K}{1-p}\cdot\frac{1}{t},
$$

where the last inequality holds by Sherman-Morrison formula. By Hoeffding bound for matrix (Lemma 2), for $x > 0$

$$\mathbb{P}\left(\left\|\mathbf{V}_t^{-1/2}\mathbf{A}_t\mathbf{V}_t^{-1/2} - \mathbf{I}_K\right\|_2 > x\right) \leq 2K\exp\left(-\frac{(1-p)^2tx^2}{8K^2}\right).$$

Setting $x = \epsilon \in (0,1)$ which will be determined later, for $t \geq 8\epsilon^{-2}(1-p)^{-2}K^2\log\frac{2Kt^2}{\delta}$ with probability at least $1 - \delta/t^2$,

$$\left\|\mathbf{I}_K - \mathbf{V}_t^{-1/2}\mathbf{A}_t\mathbf{V}_t^{-1/2}\right\|_2 \leq \epsilon,$$

$\square$

### D.3 An error bound for the Lasso estimator

**Lemma 3** (An error bound for the Lasso estimator with unrestricted minimum eigenvalue). *Let $\{\mathbf{x}_\tau\}_{\tau\in[t]}$ denote the covariates in $[-1,1]^d$ and $y_\tau = \mathbf{x}_\tau^\top\bar{\mathbf{w}} + e_\tau$ for some $\bar{\mathbf{w}} \in \mathbb{R}^d$ and $e_\tau \in \mathbb{R}$. For $\lambda > 0$, let*

$$\widehat{\mathbf{w}}_t = \operatorname*{argmin}_{\mathbf{w}} \sum_{\tau=1}^t \left(y_\tau - \mathbf{x}_\tau^\top\mathbf{w}\right)^2 + \lambda\|\mathbf{w}\|_1.$$

*Let $\bar{\mathcal{S}} := \{i \in [d] : \bar{\mathbf{w}}(i) \neq 0\}$ and $\mathbf{\Sigma}_t := \sum_{\tau=1}^t \mathbf{x}_\tau\mathbf{x}_\tau^\top$. Suppose $\mathbf{\Sigma}_t$ has positive minimum eigenvalue and $\|\sum_{\tau=1}^t e_\tau\mathbf{x}_\tau\|_\infty \leq \lambda/2$. Then,*

$$\|\widehat{\mathbf{w}}_t - \bar{\mathbf{w}}\|_{\mathbf{\Sigma}_t} \leq \frac{2\lambda\sqrt{|\bar{\mathcal{S}}|}}{\sqrt{\lambda_{\min}(\mathbf{\Sigma}_t)}}.$$

*Proof.* The proof is similar to that of Lemma B.4 in (Kim et al., 2024), but we provide a new proof for the (unrestricted) minimum eigenvalue condition. Let $\mathbf{X}_t^\top := (\mathbf{x}_1,\ldots,\mathbf{x}_t) \in [-1,1]^{d\times t}$ and $\mathbf{e}_t^\top := (e_1,\ldots,e_t) \in \mathbb{R}^t$. We write $\mathbf{X}_t(j)$ and $\widehat{\mathbf{w}}_t(j)$ as the $j$-th column of $\mathbf{X}_t$ and $j$-th entry of $\widehat{\mathbf{w}}_t$, respectively. By definition of $\widehat{\mathbf{w}}_t$,

$$\|\mathbf{X}_t(\bar{\mathbf{w}} - \widehat{\mathbf{w}}_t) + \mathbf{e}_t\|_2^2 + \lambda\|\widehat{\mathbf{w}}_t\|_1 \leq \|\mathbf{e}_t^{(j)}\|_2^2 + \lambda\|\bar{\mathbf{w}}\|_1,$$

which implies

$$\begin{aligned}
\|\mathbf{X}_t(\bar{\mathbf{w}} - \widehat{\mathbf{w}}_t)\|_2^2 + \lambda\|\widehat{\mathbf{w}}_t\|_1 &\leq 2(\widehat{\mathbf{w}}_t - \bar{\mathbf{w}})^\top\mathbf{X}_t^\top\mathbf{e}_t + \lambda\|\bar{\mathbf{w}}\|_1 \\
&\leq 2\|\widehat{\mathbf{w}}_t - \bar{\mathbf{w}}\|_1\|\mathbf{X}_t^\top\mathbf{e}_t\|_\infty + \lambda\|\bar{\mathbf{w}}\|_1 \\
&\leq \lambda\|\widehat{\mathbf{w}}_t - \bar{\mathbf{w}}\|_1 + \lambda\|\bar{\mathbf{w}}\|_1,
\end{aligned}$$

where the last inequality uses the bound on $\lambda$. On the left hand side, by triangle inequality,

$$\begin{aligned}
\|\widehat{\mathbf{w}}_t\|_1 &= \sum_{i\in\bar{\mathcal{S}}}|\widehat{\mathbf{w}}_t(i)| + \sum_{i\in[d]\setminus\bar{\mathcal{S}}}|\widehat{\mathbf{w}}_t(i)| \\
&\geq \sum_{i\in\bar{\mathcal{S}}}|\widehat{\mathbf{w}}_t(i)| - \sum_{i\in\mathcal{S}_\star}|\widehat{\mathbf{w}}_t(i) - \bar{\mathbf{w}}(i)| + \sum_{i\in[d]\setminus\bar{\mathcal{S}}}|\bar{\mathbf{w}}(i)| \\
&= \|\bar{\mathbf{w}}\|_1 - \sum_{i\in\bar{\mathcal{S}}}|\widehat{\mathbf{w}}_t(i) - \bar{\mathbf{w}}(i)| + \sum_{i\in[d]\setminus\bar{\mathcal{S}}}|\widehat{\mathbf{w}}_t(i)|
\end{aligned}$$

and for the right-hand side,

$$\|\widehat{\mathbf{w}}_t - \bar{\mathbf{w}}\|_1 = \sum_{i\in\bar{\mathcal{S}}}|\widehat{\mathbf{w}}_t(i) - \bar{\mathbf{w}}(i)| + \sum_{i\in[d]\setminus\bar{\mathcal{S}}}|\widehat{\mathbf{w}}_t(i)|.$$

Plugging in both sides and rearranging the terms,

$$\|\mathbf{X}_t(\bar{\mathbf{w}} - \widehat{\mathbf{w}}_t)\|_2^2 \leq 2\lambda\sum_{i\in\bar{\mathcal{S}}}|\widehat{\mathbf{w}}_t(i) - \bar{\mathbf{w}}(i)|. \tag{22}$$

Because $\mathbf{X}_t^\top \mathbf{X}_t$ is positive definite,

$$\|\mathbf{X}_t\left(\bar{\mathbf{w}} - \widehat{\mathbf{w}}_t\right)\|_2^2 \geq \lambda_{\min}(\mathbf{X}_t^\top \mathbf{X}_t) \sum_{i \in \bar{\mathcal{S}}} |\widehat{\mathbf{w}}_t(i) - \bar{\mathbf{w}}(i)|^2$$

$$\geq \frac{\lambda_{\min}(\mathbf{X}_t^\top \mathbf{X}_t)}{|\bar{\mathcal{S}}|} \left( \sum_{i \in \bar{\mathcal{S}}} |\widehat{\mathbf{w}}_t(i) - \bar{\mathbf{w}}(i)| \right)^2,$$

where the last inequality holds by Cauchy-Schwarz inequality. Plugging in Eq. (22) gives,

$$\|\mathbf{X}_t\left(\bar{\mathbf{w}} - \widehat{\mathbf{w}}_t\right)\|_2^2 \leq 2\lambda \sum_{i \in \bar{\mathcal{S}}} |\widehat{\mathbf{w}}_t(i) - \bar{\mathbf{w}}(i)|$$

$$\leq 2\lambda \sqrt{\frac{|\bar{\mathcal{S}}|}{\lambda_{\min}(\boldsymbol{\Sigma}_t)}} \|\mathbf{X}_t\left(\bar{\mathbf{w}} - \widehat{\mathbf{w}}_t\right)\|_2$$

$$\leq \frac{2\lambda^2 |\bar{\mathcal{S}}|}{\lambda_{\min}(\boldsymbol{\Sigma}_t)} + \frac{1}{2} \|\mathbf{X}_t\left(\bar{\mathbf{w}} - \widehat{\mathbf{w}}_t\right)\|_2^2,$$

where the last inequality uses $ab \leq a^2/2 + b^2/2$. Rearranging the terms,

$$\|\mathbf{X}_t\left(\bar{\mathbf{w}} - \widehat{\mathbf{w}}_t\right)\|_2^2 \leq \frac{4\lambda^2 |\bar{\mathcal{S}}|}{\lambda_{\min}(\boldsymbol{\Sigma}_t)},$$

which proves the result. $\qquad\square$

### D.4 EIGENVALUE BOUNDS FOR THE GRAM MATRIX.

**Lemma 4.** *For $a \in [K]$, let $\widetilde{\mathbf{x}}_a := [\mathbf{x}_a^\top, \mathbf{e}_a^\top \mathbf{p}_1, \cdots, \mathbf{e}_a^\top \mathbf{p}_{K-d}]^\top \in \mathbb{R}^d$ denote augmented features. Then, an eigenvalue of $\sum_{a \in [K]} \widetilde{\mathbf{x}}_a \widetilde{\mathbf{x}}_a^\top$ is in the following interval*

$$\left[\min\left\{\lambda_{\min}\left(\sum_{a \in [k]} \mathbf{x}_a \mathbf{x}_a^\top\right), 1\right\}, \max\left\{\lambda_{\max}\left(\sum_{a \in [K]} \mathbf{x}_a \mathbf{x}_a^\top\right), 1\right\}\right].$$

*Proof.* Let $\mathbf{P} := (\mathbf{p}_1, \ldots, \mathbf{p}_{K-d}) \in \mathbb{R}^{K \times (K-d)}$. Because the columns in $\mathbf{P}$ are orthogonal each other and to $\mathbf{x}_1, \ldots, \mathbf{x}_K$,

$$\sum_{a \in [K]} \widetilde{\mathbf{x}}_a \widetilde{\mathbf{x}}_a^\top = \begin{bmatrix} \sum_{a \in [K]} \mathbf{x}_a \mathbf{x}_a^\top & \sum_{a \in [K]} \mathbf{x}_a \mathbf{e}_a^\top \mathbf{P} \\ \sum_{a \in [K]} \mathbf{P}^\top \mathbf{e}_a \mathbf{x}_a^\top & \mathbf{P}^\top \mathbf{P} \end{bmatrix}$$

$$= \begin{bmatrix} \sum_{a \in [K]} \mathbf{x}_a \mathbf{x}_a^\top & \sum_{a \in [K]} \mathbf{x}_a \mathbf{e}_a^\top \mathbf{P} \\ \sum_{a \in [K]} \mathbf{P}^\top \mathbf{e}_a \mathbf{x}_a^\top & I_{K-d} \end{bmatrix}$$

$$= \begin{bmatrix} \sum_{a \in [K]} \mathbf{x}_a \mathbf{x}_a^\top & \sum_{a \in [K]} \mathbf{X} \mathbf{e}_a \mathbf{e}_a^\top \mathbf{P} \\ \sum_{a \in [K]} \mathbf{P}^\top \mathbf{e}_a \mathbf{e}_a^\top \mathbf{X} & I_{K-d} \end{bmatrix}$$

$$= \begin{bmatrix} \sum_{a \in [K]} \mathbf{x}_a \mathbf{x}_a^\top & \mathbf{X}\mathbf{P} \\ \mathbf{P}^\top \mathbf{X}^\top & I_{K-d} \end{bmatrix}$$

$$= \begin{bmatrix} \sum_{a \in [K]} \mathbf{x}_a \mathbf{x}_a^\top & \mathbf{O} \\ \mathbf{O} & I_{K-d} \end{bmatrix}.$$

Thus, for any $\lambda \in \mathbb{R}$, $\det(\sum_{a \in [K]} \widetilde{\mathbf{x}}_a \widetilde{\mathbf{x}}_a^\top - \lambda \mathbf{I}_K) = \det(\sum_{a \in [K]} \mathbf{x}_a \mathbf{x}_a^\top - \lambda \mathbf{I}_d)(1 - \lambda)^{K-d}$. Solving $\det(\sum_{a \in [K]} \mathbf{x}_a \mathbf{x}_a^\top - \lambda \mathbf{I}_d)(1 - \lambda)^{K-d} = 0$ gives the eigenvalues and the lemma is proved. $\qquad\square$

