# OpenReview forum: "Linear Bandits with Partially Observable Features"
_ICLR.cc/2025/Conference — Submitted to ICLR 2025_

### Official Review · Reviewer_Shzc · 2024-10-30

**Soundness:** 1
**Presentation:** 3
**Contribution:** 2
**Rating:** 3
**Confidence:** 3

**Summary:**

The paper studied linear bandits where the feature vectors of the actions are partially observable, specifically in the setting where there are no additional assumptions about the unobserved features. The authors observed that the reward of all actions lies in a subspace of $d$ dimensions, allowing for an assignment of the hidden vectors such that the corresponding unknown parameter $\mu_*$ is $d$-sparse. Based on this insight, the authors proposed a bandit algorithm that employs the Lasso estimator to leverage this sparsity. They showed that their algorithm achieves a regret bound of order $\sqrt{\tilde \sigma_{\min}^{-1} T d \log K}$, where $\tilde \sigma_{\min}$ denotes the minimum eigenvalue of the Gram matrix of the assigned feature vectors. Experiments are included to demonstrate the practical performance of the algorithm.

**Strengths:**

I think linear bandits with partially observed features is an interesting topic to study.

**Weaknesses:**

I feel there are critical theoretical flaws in the paper.

1. Even if $(I_K - P_X)U^{\top} \theta_*^{(u)}$ lies in a subspace of dimension $d$, it does not imply that $\mu_*^{(u)}$ is $d$-sparse. The sparsity only holds when the basis vectors $b_i$ align with the row space of the matrix. It seems to me that $b_i$ is arbitrarily chosen, so I do not think this claim generally holds.

2. The regret bound of the algorithm has a polynomial dependence on $\tilde \sigma_{\min}^{-1}$. However, since $\tilde \sigma_{\min}^{-1}$ is the minimum eigenvalue of the Gram matrix of a $K \times d$ matrix, it might still scale as $K$. Although the authors claim that the minimum eigenvalue is not always $O(1/K)$ because they assume $\|\tilde x_a\|_\infty \leq 1$ instead of $\|\tilde x_a\|_2 \leq 1$, I still cannot see the correspondence.

3. In Theorem 2, it is stated that $\mathcal{E}_t \subseteq [t]$, and it is also assumed that $|\mathcal{E}_t| \geq t$. I do not see how both conditions can hold unless $\mathcal{E}_t = [t]$.

**Questions:**

Please refer to the Weaknesses section above.

---

> ### Author Response · Authors · 2024-11-19
>
> 1. **On sparsity**: Your comment that "The sparsity only holds when the basis vectors $b_i$ align with the row space of the matrix" is incorrect.  The sparsity holds for any basis vectors $\{b_1, \ldots, b_{K-d}\}$ in $\mathrm{R}(\\mathbf{X})^\perp$.  By definition, if $(I_K - P_X)U^\top\theta_\star^{(u)}$ lies in a subspace of dimension $d_h$, it can be expressed as a linear combination of at most $d_h$ nonzero coefficients for any set of basis vectors $\{b_1, \ldots, b_{K-d}\}$.  While the specific coefficients $\mu_{\star,1}^{(u)}, \ldots, \mu_{\star,K-d}^{(u)}$ may vary depending on the choice of basis vectors, the sparsity itself (i.e., the presence of only $d_h$ nonzero entries) remains consistent across all bases.
>
> 2. **On the dependency on $\tilde{\sigma}_{\min}$**:  We are glad to present an updated version of our work, which includes a slight modification to the estimator.  This adjustment eliminates the dependency on $\tilde{\sigma}\_{\min}$ (and $\tilde{\sigma}\_{\max}$) in the analysis (see Theorem 3 and Appendix B.3 for details).  As a result, the dependency on $\tilde{\sigma}_{\min}^{-1}$ is no longer a concern.  This improvement is possible by normalizing the estimator with the Gram matrix as in Eq. (9) and Eq. (12).
>
> 3. **On the number of exploration rounds**: Thank you for pointing out the typo, and we have corrected $|\mathcal{E}_t| \le t$ to $|\mathcal{E}_t| \ge (\sqrt{16K} + 1)^2 2(1-p)^{-2} K^2 \log(2Kt^2/\delta)$.

---

> ### Comment · Reviewer_Shzc · 2024-11-25
>
> For [1], I still don't think it is correct in general. You should provide a proof if you believe it is correct in this scenario. Consider the following example: The vector $[1, 1, \dots, 1]^\top$ lies in the subspace spanned by the vector $[1, 1, \dots, 1]^\top$, which has dimension $1$. However, with respect to the standard basis $e_1, \dots, e_d$, you need $d$ non-zero coefficients to represent this vector.

---

> ### Author Response · Authors · 2024-11-25
>
> Dear Reviewer,
>
> We sincerely assure the reviewer that our results are correct. We appreciate the opportunity to clarify this point. Upon careful consideration, we find that the example provided in the comment does not serve as a counterexample to the results in our paper.
>
> It is important to note that $d_h$ nonzero coefficients are needed with a basis constructed **within the subspace** $\mathrm{R}(\mathbf{X})^\perp$, **NOT** within the entire space $\mathbb{R}^K$. (Note that we can compute the subspace $\mathrm{R}(\mathbf{X})^\perp$ and we find the basis $\{\mathbf{b}\_1,\ldots,\mathbf{b}\_{K-d}\}$ within this subspace, rather than using the full space $\mathbb{R}^K$.) In your example, our algorithm identifies the basis in the subspace **spanned** by the vector $[1,1,\ldots,1]^\top$ but not the Euclidean basis in $\mathbb{R}^d$. As a result, our basis is $\mathbf{b}_1 = [1, 1, \ldots, 1]^\top$, meaning only one basis vector is required to represent $[1, 1, \ldots, 1]^\top$.
>
> We are happy to  make this distinction clearer in the revised manuscript to ensure there is no ambiguity. We sincerely hope that our explanation has addressed your concerns. Given the correctness of our results, we respectfully request that you reconsider your evaluation of our work in light of this clarification.
>
> Thank you again for your time and consideration.

---

> ### Comment · Reviewer_Shzc · 2024-11-25
>
> Let us assume $K =  n + 1$ and $X = [0, \dots, 0, 1]^\top$. In this case, $e_1, \dots, e_n$ form a feasible basis for the subspace $R(X)^\perp$. Furthermore, let us assume $U = [[1, \dots, 1, 0]]$ and $\theta_*^{(u)} = [1]$. Then, we have $(I_K - P_X) U^\top \theta_*^{(u)} = [1, \dots, 1, 0]^\top$. It is clear that $d_h := \mathrm{dim} (R((I_K - P_X) U^\top) ) = 1$. However, to express $(I_K - P_X) U^\top \theta_*^{(u)}$ under the Euclidean basis, $n$ non-zero coefficients are required to represent this vector.

---

> > ### Author Response · Authors · 2024-11-26
> >
> > We appreciate the continuing discussion to assure the correctness of our results. Again, your example cannot be a counter example of our results since the dimension of $\mathbf{X}$ does not match our setting. While $\mathbf{X}$ in your example is in $\mathbb{R}^{1 \times K}$, our $\mathbf{X}$ must be in $\mathbb{R}^{d \times K}$. In order for your example to fit in our setting, it is required that $d=1$ and $K=2$, which forms $\mathbf{e}\_1$ as a feasible basis for the subspace $\mathrm{R}(\mathbf{X})^\perp$. Then, we have $(I\_K-P\_X)U^\top \theta\_{\star}^{(u)} = [1,0]^\top$ and which is exactly $\mathbf{e}\_1$, and we can clearly express it with only $d_h=1$ vector. We believe it is impossible to find a counter example against finding the $d_h$ basis for the subspace $\mathrm{R}((I_K-P_X)\mathbf{U})$ since the dimension of the subspace is $d_h$.

---

> > > ### Comment · Reviewer_Shzc · 2024-12-01
> > >
> > > I just have one question regarding this. Why is it required to have the number of actions set as $K = 2$? For instance, why can we not have $K = 11$ arms while the dimension is $d=1$, so that $X = [0, 0, 0, 0, 0, 0, 0, 0, 0, 0, 1]$?

---

> > > > ### Author Response · Authors · 2024-12-02
> > > >
> > > > Thank you, and we are happy to answer your question.  To begin, we’d like to emphasize that in our general problem setting, we assume $K \geq 1$.  The case where $K = 2$ is specific to the scenario you described in your earlier comment.
> > > >
> > > > Regarding your example with $d = 1$ and $K = 11$, this does not align with our setting because the feature vectors in your example are repetitive.  Specifically, the entire feature matrix is represented as $[X; U] = [(0, \cdots, 0, 1); (1, \cdots, 1, 0)]$, where the semicolon (;) separates the rows of the matrix. This structure effectively reduces the problem to a $K = 2$ armed linear bandit problem.
> > > >
> > > > To relieve your concerns further, we propose a new definition of the dimension of the row space of $(\mathbf{I}\_{K} - P_{\mathbf{X}})\mathbf{U}^\top$, adaptive to the selected basis:  $\tilde{d}\_h = \min\{h \geq 1, \{i_1,\ldots,i\_h\} \subseteq [K-d] : \mathrm{R}(\mathbf{U}) \cap \mathrm{R}(\mathbf{X})^\perp \subseteq \mathrm{R}(\mathbf{b}\_{i_1}^{\top},\ldots,\mathbf{b}\_{i_h}^{\top})\}.$
> > > >
> > > > Note that $\tilde{d}_h$ still ranges from $0$ to $K-d$ and importantly, the algorithm does not require prior knowledge of $\tilde{d}_h$.  Additionally, no modifications are needed for the remaining analysis, and our main contribution is that we achieve regret sublinear in $T$ for linear bandits with partially observable features, which is more challenging than misspecified linear bandits.
> > > >
> > > > We sincerely hope that this clarifies any remaining questions.

---

### Official Review · Reviewer_ezt5 · 2024-11-03

**Soundness:** 3
**Presentation:** 3
**Contribution:** 3
**Rating:** 6
**Confidence:** 3

**Summary:**

The paper studies a novel linear contextual bandits problem, which differs from the standard finite-arm linear contextual bandits problem in the following way.
1. There are $K$ arms in total and the contextual vector of each arm is fixed through horizon $T$.
2. Among all dimensions $d$, there are a few of them are not observed by the player.
3. The contextual vector has bounded $\ell_\infty$-norm and the parameter $\theta^\star$ has bounded $\ell_1$-norm.

The paper presented an algorithm for the problem, gave matching lower and upper bounds up to logarithmic factors, and demonstrated the effectiveness of the paper in numerical experiments.

**Strengths:**

1. The paper gave a rather complete study on the proposed problem. As mentioned before, the paper included algorithm designs, proofs to the lower and upper bounds, and numerical experiments.
2. The lower and upper bounds are matching up to log factor.

**Weaknesses:**

1. The paper made a rather strong assumption that $(\mu,\theta)$ are bounded in $(\ell_\infty, \ell_1)$ which is different from the standard $(\ell_2, \ell_2)$ setup. This limits the comparison of this work over previous ones.

**Questions:**

1. Line 1325: "Cauchy-Schwartz" should be "Cauchy-Schwarz".
3. Is the observed part of the feature space really used? What would happen if we just pretend all features are not observable and apply the algorithm, i.e. replace $d,d_h$ by $d'=0$ and $d'_h=d+d_h$? It confused me a lot that the regret bound only depends on $d+d_h$.
2. I suggest the authors make more explicit about the regret bound's dependency on $\widetilde \sigma_{\min}^{-1}$ to avoid confusion.
3. At Line 475-480, I don't see why $\widetilde \sigma_{\min}$ is constant.

---

> ### Author Response · Authors · 2024-11-19
>
> [W1] **On the boundedness assumption**: We are pleased to share that, by removing the dependency of our regret bound on $\tilde{\sigma}_{\min}$, we have also eliminated the requirement for the bounded feature and parameter assumption in our problem setting. We kindly invite you to review the updated version of our paper for further details.
>
> ---
>
> Answers to questions:
>
> [Q1] **On the typo in Line 1325**: Thank you for bringing the typo to our attention; we have now corrected it.
>
> [Q2]  **On the utilization of observable features**: The observable part is utilized to identify $\mathrm{R}(\mathbf{X})^{\perp}$ and construct the basis vectors accordingly. Without leveraging the observed contexts, the algorithm would rely on arbitrary basis vectors, resulting in a regret of $\tilde{O}(\sqrt{K T})$.
>
> [Q3-4]  **On the dependency on $\tilde{\sigma}_{\min}$**:  As mentioned in the first response, we have successfully removed the dependency on $\tilde{\sigma}_{\min}$ from our regret bound by normalizing the coefficients for the lasso estimator with the Gram matrix slightly as in Eq. (9) and Eq. (12).

---

> > ### Comment · Reviewer_ezt5 · 2024-11-24
> >
> > For [Q2], when there's no observable part, I thought we simply have $R(X)^\perp = \mathbb R^d$ and the basis can be the standard basis of the space. Could you explain more on why arbitrary basis lead to $\widetilde O(\sqrt{KT})$?

---

> ### Author Response · Authors · 2024-11-25
>
> Thank you for your question. We are happy to address it. When there is no observable part, we have $\mathrm{R}(\mathbf{X})^\perp = \mathbb{R}^K$, not $\mathbb{R}^d$, since we are considering the row space of features. Then our proposed algorithm uses the standard basis in $\mathbb{R}^K$ and the regret bound leads to $\tilde{O}(\sqrt{KT})$. We hope this clarifies your question. We sincerely hope that our explanation has clarified your concerns. In light of this clarification, we respectfully request that you reconsider your evaluation of our work.
>
> Thank you again for your time and consideration.

---

> ### Author Response · Authors · 2024-11-30
>
> Thank you for your question.
> To better address your question, could you please clarify what you mean by "just assign standard basis to the arms" and "select $d$ out of $K$ arms"? A more specific explanation of your question would help us provide a precise and relevant response.

---

> > ### Author Response · Authors · 2024-12-04
> >
> > Dear Reviewer ezt5,
> > We sincerely thank you for taking the time to review our paper and for considering our clarifications during the discussion. We also noticed that you have increased your score, and we appreciate your recognition of the value and contributions of our work.

---

### Official Review · Reviewer_rtzC · 2024-11-04

**Soundness:** 1
**Presentation:** 3
**Contribution:** 1
**Rating:** 3
**Confidence:** 4

**Summary:**

This paper consider a bandit setting with a fixed set of arms and with unobservable features. The authors propose to augment the feature space with basis vectors and propose algorithms to provide sublinear regret guarantee.

**Strengths:**

Designing algorithms to tackles unobservable confounding factors is important for bandit problems.

**Weaknesses:**

1. Assumption 1 requires features of all arms to be fixed over time, which is not common and very strong. I don't think this assumption matches practices. Besides, I didn't see how this can be "slightly modified" into a time-varying feature setting as claimed by the authors. If it's simple, please use the time-varying feature setting as the main setting, which is common in the bandit literature.
2. The lower bound analysis doesn't seem right to me. The last step relies on the argument that the agent cannot identify $x_{a_*}$ from $x_{a'}$. However, the observed outcome $y$ for these two arms $a_*$ and $a'$ will be different since $\langle z_{a_*}, \theta_*\rangle \ne $\langle z_{a'}, \theta_*\rangle$, even though $x_{a_*}=x_{a'}$. Therefore, a linear regression will be able to distinguish these two actions. Intuitively, even if there are unobservable features, linear regression will still be able to tell one action from others since the unobservable values will be projected into the observable feature space. Thus, it's hard to believe the lower bound will scale as T.
3. The coupling steps require more explanations and intuition as well. The authors only argue adaptation of the existing DR method is challenging but didn't explain the intuition. Why coupling can help with the analysis when arms are fixed?
4. The comparison in the experiments does not seem fair. The benchmarks in the experiments are not of the same type of bandit algorithm with the proposed ones. I would suggest the authors at least add DR Lasso Bandit (Kim & Paik, 2019).

**Questions:**

See my comments above.

---

> ### Author Response · Authors · 2024-11-19
>
> 1. **Fixed contexts assumptions are common in the parametric bandit literature**:  The parametric bandit literature includes several studies where the reward is modeled as a function of each action and an unknown parameter, with the action set fixed throughout the learning horizon—an approach that aligns with the problem setting of our paper.  For instance, Abbasi-Yadkori (2009) investigates a linear bandit algorithm that employs forced sampling, achieving a regret bound of $O(d\sqrt{T})$ under a fixed set of observable features.  Similarly, Filippi et al. (2010) analyze a stochastic bandit problem with a generalized linear model (GLM) reward structure, also considering fixed observable features.  Additionally, this assumption has been applied to address practical problems, including model selection (Zhu \& Nowak, 2022), asymptotic analysis (Lattimore \& Szepesvari, 2017), and the development of experimental design-based algorithms (Wakenmaker et al., 2021), fairness-focused bandit algorithms (Chen et al., 2020), and bandit algorithms under safety constraints (Amani et al., 2019).  These examples illustrate that the assumption imposed in our paper is widely adopted and holds significant relevance in practical applications.
>
> **Slightly modified analysis for time-variant contexts**:  Under the assumption that $\tilde{\sigma}\_{\min} := \min\_{t \in [T]} \lambda\_{\min} \left( \sum_{a \in [K]} \mathbf{x}\_{a,t} \mathbf{x}\_{a,t}^\top \right) > 0$ for deterministic contexts, or $\tilde{\sigma}\_{\min} := \min\_{t \in [T]} \lambda\_{\min} \left( \mathbb{E} \left[ \sum\_{a \in [K]} \mathbf{x}\_{a,t} \mathbf{x}\_{a,t}^\top \right] \right) > 0$ for stochastic contexts, repeatedly applying feature vector augmentation to $\mathbf{X}\_t := (\mathbf{x}\_{1,t}, \ldots, \mathbf{x}\_{K,t})$ for time-variant features $\mathbf{x}\_{a,t}$ produces the same theoretical results.  While this setting accommodates general cases in bandit problems, for simplicity, we will address this in the Appendix.  Note that the unobserved features $\mathbf{u}\_a$ must remain fixed over time; otherwise, the problem becomes so challenging that achieving sublinear regret with any algorithm is infeasible.
>
> 2. **On the lower bound**:  Your comment "a linear regression will be able to distinguish these two actions" is incorrect.  For any linear regression estimator $\widehat{\theta}$ for $\theta_{\star}$, the reward estimate for the two actions, $\mathbf{x}\_{a_\star}^\top \widehat{\theta}$ and $\mathbf{x}\_{a^\prime}^\top \widehat{\theta}$, are indistinguishable when $\mathbf{x}\_{a_\star}=\mathbf{x}\_{a^\prime}$, although the observed rewards, $y_{a_\star}$ and $y_{a^\prime}$, are distinguishable (note that we cannot use $\mathbf{z}\_{a_\star}^\top \widehat{\theta}$ and $\mathbf{z}\_{a^\prime}^\top \widehat{\theta}$ because of partial observability).  Since our lower bound (Theorem 1) applies only to algorithms that use observed contexts (excluding multi-armed bandit algorithms that ignore the contexts), the algorithms cannot distinguish the two actions and suffer regret linear in the time horizon $T$.
>
> 3. **How coupling eases the adaption of DR**:  Coupling replaces the $\epsilon_t$-greedy policy with a multinomial distribution $\phi_{1,t}, \ldots, \phi_{K,t}$.  When we use DR estimation with the $\epsilon_t$-greedy policy, the inverse probability $\epsilon_t^{-1} := \sqrt{t}$ appears in the pseudo-reward (10), causing the variance of the pseudo-reward to explode.  Therefore, we couple the $\epsilon_t$-greedy policy with the multinomial distribution (9) to bound the inverse probability weight $\phi_{a,t}^{-1} = O(K)$. We have included this in the revised manuscript.
>
> 4. **On comparison with DR Lasso Bandit**: We have already included the DR Lasso Bandit (Kim \& Paik, 2019) in our comparison experiments, as clearly presented in Figures 2, 3, and 4 of the original manuscript.
>
> **References for first question**
> - Yasin Abbasi-Yadkori. Forced-exploration based algorithms for playing in stochastic linear bandits. 2009.
> - Sanae Amani et al.,  Linear stochastic bandits under
> safety constraints. 2019.
> - Yifang Chen et al.,
> Fair contextual multi-armed bandits: Theory and experiments. 2020.
> - Sarah Filippi et al., Parametric bandits: The
> generalized linear case. 2010.
> - Tor Lattimore and Csaba Szepesvari. The End of Optimism? An Asymptotic Analysis of Finite-Armed
> Linear Bandits. 2017.
> - Andrew Wagenmaker et al., Experimental design for regret
> minimization in linear bandits. 2021.
> - Yinglun Zhu and Robert Nowak. Pareto optimal model selection in linear bandits. 2022.

---

> ### Comment · Reviewer_rtzC · 2024-11-19
> **Response**
>
> 1. Unfortunately, Abbasi-Yadkori (2009) does not require a fixed set of feature vectors. The set of feature vectors is arbitrary over time. If I understand correctly, you require the set of features to be the same over time. I don't  think this is common. Some of the paper you listed do not even satisfy this requirement.
> 2. If I pull $x_{a*}$ and $x_{a'}$. the outcome $y$'s are different, why cannot I identify that these two arms are different and have different unobservable features?
> 3. Please add the explanations to the papers in more detail.

---

> ### Author Response · Authors · 2024-11-20
>
> # Response for the first comment
>
> Thank you for your comment.
> At face value, the problem setup in Abbasi-Yadkori (2009) considers a fixed arm set. However, we acknowledge that their framework may also allow for time-varying features, as the reviewer has pointed out. While we of course recognize that allowing the feature set to vary over time is standard (that is not an issue at all) in conventional linear contextual bandit problems, we sincerely hope that the reviewer and we can agree on the following common grounds:
>
> **(1) Fixed arm sets are NOT uncommon in variants of linear bandit problems: in fact, natural with partial observability.**
>
> In many variants of linear bandit problems, assuming a fixed arm set is rather common (not uncommon), particularly in linear bandit settings with misspecification (which can be formulated as linear bandits with partially observable features). For instance:
> - Ghosh et al. (2017) and Lattimore et al. (2020) assume fixed contexts over time in their analysis of linear bandits with misspecification.
> - Other works, such as Park \& Faradonbeh (2024) and Kim et al. (2023), consider stochastic contexts (e.g., Gaussian-distributed features), which are time-varying but impose additional assumptions on the latent features. Specifically:
>   - Park \& Faradonbeh (2024) assumes latent features have a mean of zero.
>   - Kim et al. (2023) assumes that observed features are linear transformations of latent features via a full-rank matrix (in expectation).
> To the best of our knowledge, **no existing work tackles linear bandits with misspecification or partially observable features under a setting where the feature vector varies arbitrarily over time** without additional assumptions on the latent features. In our work, we assume fixed features but make no structural assumptions between observed and latent features, which justifies the fixed arm set setting.
>
> **(2) A fixed arm set is not necessarily easier (in general).**
>
> Even for the standard linear contextual bandit (note our problem is with partial observability: see Part (3) below), allowing a time-varying arm set does NOT inherently make the problem more challenging. On the contrary, assuming non-fixed, stochastic contexts (e.g., multivariate Gaussian features) can rather simplify and ease the problem, as shown in prior work:
> - Kannan et al. (2018) and Bastani et al. (2021) demonstrate that stochastic contexts enable sharper regret bounds, with greedy policies achieving sub-linear regret performance.
>
> In fact, these **non-fixed stochastic contexts are often considered strong assumptions** because they simplify the problem significantly. Thus, we respectfully object to the reviewer’s assertion that fixed features represent a “very strong” assumption.
> If anything, **there is no inherent "strong-ness" in assuming fixed features, as it neither simplifies the problem nor leads to better results**. If the reviewer remains unconvinced, we encourage them to assume fixed contexts in the problem of Abbasi-Yadkori et al. (2011) (or any typical linear bandit analysis), work out the analysis, and examine the outcomes. Does it yield a sharper regret bound? Does it simplify the analysis? The answer is unequivocally, NO. Hence, a fixed feature set is not a strong assumption by any theoretical standard.
>
> **(3) Whether features are fixed or varying should not be grounds for rejection.**
>
> Most importantly, the choice between fixed and varying features alone is not a matter of strong-ness but a matter of problem suitability. Our problem setting involves unobservable portions of each feature vector. **If these latent features were to change arbitrarily over time, the problem itself would become non-learnable**, making the problem ill-posed. Hence, given this, assuming fixed features is natural and justified.
>
> In fact, **our problem setting with a fixed arm set and partial observability is much harder** than the typical linear bandit settings with full observability, whether the features are fixed or even time-varying.
> The presence of unobservable features introduces significant complexity. This is not a matter of opinion but rather a straightforward fact, and we sincerely hope that we can agree on this simple premise.
>
> We firmly believe that our contributions under this challenging setting warrant recognition rather than rejection based solely on this assumption.
>
> For **time-varying observed features**, we have included an algorithm in Appendix B that achieves $\tilde{O}(\sqrt{(d+K)T})$ regret bound.
>
> ---
> ### References
> - Ghosh et al. (2017), "Misspecified Linear Bandits"
> - Kannan et al. (2018), *A Smoothed Analysis of the Greedy Algorithm for the Linear Contextual Bandit Problem*
> - Bastani et al. (2021), *Mostly Exploration-Free Algorithms for Contextual Bandits*
> - Kim et al. (2023), "Contextual Linear Bandits Under Noisy Features: Towards bayesian oracles"
> - Park \& Faradonbeh (2024), "Thompson sampling in partially observable contextual bandits"

---

> ### Author Response · Authors · 2024-11-20
>
> # Response for the second comment
>
> Note that the observed $y\_a = \langle \mathbf{z}\_{a}, \theta\_{\star} \rangle + \eta\_a$ includes the measurement error. Thus, although the observed $y$ is different, we do not know whether they have different expected values or not. With a sufficient number of reward samples, some MAB algorithms that do not depend on the features may distinguish $a_{\star}$ and $a^\prime$, but our lower bound does not consider those algorithms. We show that $a_{\star}$ and $a^{\prime}$ are indistinguishable for the algorithms whose policies operate on the observed features $\tilde{\mathbf{x}}\_a$ only. The lower bound is not about all algorithms. One example is the LinUCB (Li et al., 2010) algorithm using $\tilde{\mathbf{x}}\_a$, which clearly does not distinguish the two arms that have the same feature vectors even though the observed rewards are different.
>
> For example, consider a specific case following the setting in Theorem 1. Let the dimension of observed features, $d$, be 3, and the total horizon, $T$, be 9. Following this setup, the observed features of each arm are calculated as follows: $\mathbf{x}\_{a\_\star} = [-1, -1, -1]^\top, \quad \mathbf{x}\_{a'} = [-1, -1, -1]^\top, \quad \text{and} \quad \mathbf{x}\_{a\_o} = [1, 1, 1]^\top$.
>
> Based on the theorem, the expected reward for each arm, incorporating unobserved features, is given by $[1/3, -1, -1/3]^\top$. Now, consider, for example, the decision rule of LinUCB using ridge regression with a regularization parameter $\lambda = 1$.  Using ridge regression, we obtain $\hat{\theta} = [1/30, 1/30, 1/30]^\top$, which results in:  $\mathbf{x}\_{a\_\star}^\top \hat{\theta} = \mathbf{x}\_{a'}^\top \hat{\theta} = -1/10$.
>
> For the uncertainty term, since the observed feature vectors for both $a\_\star$ and $a'$ are identical, their norms (even when weighted by the Gram matrix) are also identical. Consequently, linear bandit algorithms that rely solely on observable features are unable to distinguish between $a\_\star$ and $a'$.
>
> # Response for the third comment
>
> We have already included the explanations of how coupling eases the adaption of DR in Section 4.2 (Page 7). We are more than happy to improve our manuscript.

---

> ### Author Response · Authors · 2024-11-25
>
> Dear Reviewer,
>
> Upon reviewing the comments in the weaknesses that the reviewer had originally mentioned (along with the follow-up comments), we **strongly believe that all the points have been addressed and clarified** as follows:
>
> 1. **On fixed features**: Please refer to the detailed discussion in the section above (titled "Response for the first comment"). To summarize, our results are as follows:
>
> | Observable Features | Unobservable Features | Learnability | Results |
> |---------------------|------------------------|--------------|---------|
> | Fixed               | Fixed                 | Yes          | $\tilde{O}(\sqrt{(d+d_h)T})$ (Section 5.2) |
> | Varying             | Fixed                 | Yes          | $\tilde{O}(\sqrt{(d+K)T})$ (Appendix B) |
> | Varying             | Varying               | No (ill-posed problem) | - |
>
> We respectfully highlight that the revised version includes results for varying observable features (Appendix B). Penalizing our work for not addressing an unlearnable problem setting (e.g., varying observable and varying unobservable features) would be unwarranted. We sincerely hope the reviewer acknowledges that this issue has been fully resolved.
>
> 2. **Lower bound**: Regarding the lower bound, we confirm its correctness. If there are still any parts that remain unclear, please do not hesitate to let us know, and we will do our best to provide additional clarification within the remaining discussion period.
>
> 3. We have included further explanations on how coupling simplifies the adaptation of DR, as requested (page 7: Section 4.2).
>
> 4. The comparison with the DR Lasso Bandit was already included in the original submission (pages 9-10: Figure 2,3,4).
>
> Given these clarifications, we sincerely believe that all points have been adequately addressed and are no longer issues—some were not issues even from the beginning. If our responses have sufficiently resolved your concerns, we kindly and respectfully request that you reconsider your assessment, reflecting the value and contributions of our work.
>
> Thank you again for your time and consideration.

---

### Official Review · Reviewer_44d2 · 2024-11-10

**Soundness:** 3
**Presentation:** 3
**Contribution:** 3
**Rating:** 8
**Confidence:** 4

**Summary:**

The authors consider a regret minimizing linear bandits with K arms. Unlike the standard linear bandit setting, they assume that each arm has a set of unseen or latent features which affect the reward. As a result, standard linear bandit algorithms (eg OFUL) can’t be applied. The authors address this issue by projecting the problem to an appropriate space where an orthogonal complement to the span of the arms is augmented, and then performing an epsilon greedy style approach in this space. Critical to the solution is the use of Doubly robust estimation procedures. They show strong empirical results.

**Strengths:**

Overall I rather enjoyed this paper and thought the approach of the authors was clever. Intuitively, in an extreme case, if you do not observe any features, you can’t hope for better than O(\sqrt(KT)) regret coming from standard MAB results. However, by projecting the feature vectors appropriately, the authors can exploit linear dependencies between the unseen components and seen components to do better.

What’s perhaps most impressive is Theorem 2, which provides an estimator that can effectively bound the estimation error using a doubly robust estimator.

**Weaknesses:**

I think there are some technical weaknesses and gaps:
a) It seems unfortunate that Theorem 2 has a burn in period that depends on K. Can the authors argue that this is necessary? (It seems so?)
b) I think some contextualization of Theorem 2 would be helpful - for example, which components of the error come from the bias of using 10 to estimate rewards? c) Forced uniform exploration seems critical to your method - do you think you could have avoided it?
D) It feels like the regret could be as bad as O(sqrt(KT)) - for example, if the seen features all have the same inner product with theta_s, and the unseen dimension encodes a basis. Can you clarify those cases?

Comments on Structure:
Overall I thought the paper was well written, but there were some gaps in exposition that would have helped. Here’s some concrete suggestions
a) Perhaps add a table contextualizing the regret in different ranges. Eg, how big can d_h get?
b) I think some concrete examples explaining your extrapolation between MAB and the linear bandit and how DR estimation helps this can help.

Finally I thought the discussion on sparsity was interesting…but I failed to understand it. d_h could be much smaller than d_u, but this does not imply sparsity?

**Questions:**

see above

---

> ### Author Response · Authors · 2024-11-19
>
> We are grateful for your invaluable feedback and acknowledgment of our contributions. We are more than happy to address each of your comments and questions.
>
> Questions
> 1. About the burn-in period: Although the $K$-dependent burn-in period is necessary in our analysis, it remains unclear whether the same order of burn-in is required for the problem itself -- our results show the sufficiency of such a burn-in period. The dependency on $K^3$ in the burn-in period arises from three components: identifying the best arm, determining the nonzero entries in the augmented parameter $\mu\_{\star} \in \mathbb{R}^K$, and accounting for the importance weight from $\phi\_{k,t}$ as defined in equation (7).  These components multiply to form the overall $K^3$ dependency.  Please note that if an upper bound $D$ for $d + d_h$ is known and $D \ll K$, it may be possible to replace the second $K$ with $D$.  For bandits with partially observable features, we believe it is hard to reduce dependency on $K$ in the burn-in period.  Nonetheless, we would like to emphasize that the burn-in period grows only by $\log T$, making it a non-leading term in the overall analysis.
>
> 2. Contextualization of Theorem 2: We included the following after Theorem 2:  "The consistency is proved by bounding the two components of the error in the pseudo-rewards defined in Eq. (11):  (i) the noise of the reward and (ii) the error of the imputation estimator ${\check{\mu}}\_{t}$. Since (i) is sub-Gaussian, it can be bounded using martingale inequalities.  For (ii), the imputation error $\tilde{x}\_a^\top (\check{\mu}^{L}\_{t} - \mu\_{\star})$ is multiplied by the mean-zero random variable $(1-\frac{\mathbb{I}(\tilde{a}\_t=a)}{\phi\_{a,t}})$ and thus it can be bounded by $\Vert \check{\mu}^L\_t - \mu\_{\star}\Vert\_1 / \sqrt{t}$."
>
> 3. On the forced uniform exploration:  Forced "uniform" exploration is not strictly essential. As long as a sufficient number of samples are collected from the $K$ arms, it is acceptable for some arms to be sampled more frequently than others.  Due to the partially observable nature of the problem, sufficient forced sampling remains necessary; however, it could be adapted to some extent.  While uniform sampling is a standard approach, there may exist more effective adaptive sampling methods for exploration.
>
> 4. Various regret bounds over different features:  When $\mathrm{R}(\mathbf{U}) = \mathbb{R}^K$, the regret could potentially be as bad as $O(\sqrt{KT})$.  However, this is not always the case, such as in scenarios where $\mathrm{R}(\mathbf{U}) \subset \mathrm{R}(\mathbf{X})$, the relationship between the observable and unobservable parts can mitigate this issue.  Specifically, if there is a linear correlation between the observable and unobservable features (as in \cite{park2022regret}), the unobservable dimensions may be inferred more effectively, achieving a $\tilde{O}(\sqrt{dT})$ regret bound.
>
> 5. On the sparsity of $d_h$:  When $d_h \ll d_u$, the unobservable component is largely linear with respect to the observable component.  Consequently, only $d_h$ basis vectors are needed to estimate the reward, rather than estimating all $d_u$ unobservable parameters.  This leads to a regret bound of $\tilde{O}(\sqrt{(d + d_h)T})$, which is significantly better than $\tilde{O}(\sqrt{(d + d_u)T})$.
>
> Suggestions
> 1. On adding a table for the range of regret: Thanks for your invaluable suggestion, and we have added a table for cases when $d_h = 0$ (linear bandit case) and $d_h = K-d$ (MAB case).
>
> 2. On examples for extrapolation between MAB and linear bandits: The DR approach uses every $K$ context vector, yielding a better gram matrix by replacing $\mathbf{x}\_{a_t}\mathbf{x}\_{a_t}^\top$ with $\sum\_{a \in [K]} \mathbf{x}\_{a}\mathbf{x}\_{a}^\top$.
> This improvement reduces the number of forced exploration rounds needed to satisfy the restrictive minimum eigenvalue condition, which is crucial for the convergence of the Lasso estimator.

---

> > ### Comment · Reviewer_44d2 · 2024-11-27
> > **Thanks for the response**
> >
> > Hi,
> >
> > Thank you for your response. It answered all my questions adequately.

---

> > > ### Author Response · Authors · 2024-12-01
> > >
> > > Thank you very much for your continued support!

---

### Public Comment · ~Sungwoo_PARK5 · 2025-02-11
**Notification of Revision Update**

We emphasize that, despite some of the concerns raised, the essence of our lower bound statement and the central arguments in the initial version remain valid. Despite the correctness of our results, to prevent any misunderstanding, we further significantly improved the scope and presentation of our lower bound (with new analysis). The rest of the revisions primarily clarify our assumptions, define the scope more rigorously, and incorporate additional analyses and comparisons to resolve any ambiguity.

A revised version of our manuscript has been posted on arXiv (http://arxiv.org/abs/2502.06142). We invite the reader to refer to the updated text, especially the new lower bound, newly added appendices, definitions, and tables. We appreciate the time and effort invested by the area chair and the reviewers, and **we strongly believe that the revisions address all points raised**.

We summarize the key improvements as follows:

1. On Clarity and Scope of the Lower Bound (Theorem 1)
    * Despite the correctness of the lower bound, some comments indicated that the presentation of the theorem may have been unclear to readers. We are happy to address this.
    * **Improvement**: We have provided a formal definition of a policy that depends solely on observed features (see Definition 1 in Appendix D). Additionally, we present novel analyses, showing that the regret bound grows linearly when unobserved features are ignored, in two cases: (1) for the well-known linear bandit algorithms `OFUL` and `LinTS` (see Appendix C.1) and (2) for policies that satisfy Definition 1.

2.  Explanation on Fixed Feature Assumption and Relaxation to Varying Observed Features
    * **Improvement**: We have introduced a setting and an algorithm that handle time-varying observed features (see Appendix B). Additionally, we have added a table (Table 2) summarizing learnability and our results across different cases, considering whether observed and unobserved features are fixed or varying. Furthermore, we have clarified that while the fixed feature assumption is standard in linear bandits with model misspecification, allowing unobserved features to change arbitrarily over time would make the problem fundamentally non-learnable (see Section 3.2).

3. On Sparsity and Justification of Introducing Lasso Estimator
    * **Improvement**: We have added a formal definition of the quantity $d\_h$, the number of nonzero coefficients that express the projection of the latent portion of reward, and have clarified how this quantity varies under different circumstances (see Section 4.1). To further improve clarity, we have also included a table summarizing the regret bounds of our algorithm depending on $d\_h\in[0, K-d]$ (see Table 1).

4. Other improvements
    * We have included the `DRLasso` (Kim \& Paik, 2019) in our comparison experiments (see Section 6).
    * We have clarified our explanation about how the coupling steps eases the adaptation of doubly robust estimator (see Section 4.2).
    * We have eliminated the dependency of our algorithm's regret bound on $\tilde{\sigma}\_{\min}$, the minimum eigenvalue of the Gram matrix of augmented feature vectors (see Theorem 3 and Appendix C.3 for proof).
    * We have corrected typos.

===========

References

* Yasin Abbasi-yadkori, Dávid Pál, and Csaba Szepesvári (2011). Improved algorithms for linear stochastic bandits.
* Shipra Agrawal and Navin Goyal (2013). Thompson sampling for contextual bandits with linear payoffs.
* Gi-Soo Kim and Myunghee Cho Paik (2019). Doubly-robust lasso bandit.

---

### Meta-Review · Area_Chair_5D3v · 2024-12-22

**Metareview:**

This papers has received reviews with high variance. A point that was raised that I found particularly problematic concerns the lower bound. It took multiple interactions between a reviewer and the authors to clarify why arms cannot be identified. The problem lies, in my opinion, in the way the theorem is phrased. It gives the impression that it is a `normal' lower bound but, in fact, it holds only for some algorithms. The phrase `that are dependent on observed features' is meant to convey this. But strictly speaking it should have been `only on observed features'. It is then unclear which algorithms  are actually covered by this lower bound since it essentially does not allow you to look at the pay-off. The sentence below the theorem also gives the impression that this is typical lower bound. All of this is very misleading and is used as a motivation for what follows. I therefore believe that the paper is not ready for publication in this current form.

**Additional Comments On Reviewer Discussion:**

There was a significant discussion between reviewers and authors. The discussions evolved around clarifying assumptions and sparsity questions, but was not fully resolved.

---

### Decision · Program_Chairs · 2025-01-22

Reject